# Cation-π interactions enabled water-stable perovskite X-ray flat mini-panel imager

Wanting Pan[1], Yuhong He[1], Weijun Li[1], Lulu Liu[1], Keke Guo[1], Jianglei Zhang[1], Chao Wang[2], Bao Li[1], Hu Huang[2], Junhu Zhang[1,3], Bai Yang[1,3] & Haotong Wei[1,3] ✉

Sensitive and stable perovskite X-ray detectors are attractive in low-dosage medical examinations. The high sensitivity, tunable chemical compositions, electronic dimensions, and low-cost raw materials make perovskites promising next-generation semiconductors. However, their ionic nature brings serious concerns about their chemical and water stability, limiting their applications in well-established technologies like crystal polishing, microprocessing, photolithography, etc. Herein we report a one-dimensional tryptamine lead iodide perovskite, which is stable in water for several months as the strong cation-π interactions between organic cations. The one-dimensional and two-dimensional tryptamine lead iodide perovskite tablets are switchable through thermal-annealing or water-soaking treatments to relax microstrains. The water-stable and microstrain-free one-dimensional perovskite tablets yield a large sensitivity of $2.5 \times 10^6$ µC $Gy_{air}^{-1}$ $cm^{-2}$ with the lowest detectable dose rate of 5 $nGy_{air}$ $s^{-1}$. Microelectrode arrays are realized by surface photolithography to construct high-performance X-ray flat mini-panels with good X-ray imaging capability, and a record spatial resolution of 17.2 lp $mm^{-1}$ is demonstrated.

Sensitive and stable X-ray detectors have been widely deployed in our daily life and military applications, for example, medical imaging, security inspection, academic research, industry product quality inspection, and so on[1,2]. However, their performances are still not satisfactory by considering the applied high doses in X-ray generators and the subsequent damage to the human body. Although scintillators are more compatible with the commercial readout system and dominate the current markets[3,4], semiconductors in direct detection mode are attractive in higher spatial resolution and lower-dosage X-ray imaging[5–7]. Traditional semiconductors such as silicon (Si) and amorphous selenium (α-Se) have limited absorption of hard X-rays, and thus result in a low sensitivity[8,9]. Large cadmium zinc telluride (CdZnTe) and high-purity germanium (Ge) single crystals are often obtained from the high-temperature Bridgeman method, and it is also difficult to control their intrinsic defects and subsequent integration

on large-area read-out circuit substrates in a low-cost way. High-purity Ge even has to be operated under a liquid nitrogen atmosphere, which greatly limits the development of X-ray detectors in direct detection mode[10,11].

Halide perovskite X-ray detectors encountered unprecedented developments in recent years due to their advantages of strong stopping power, large µτE product (charge carrier mobility (µ), charge carrier lifetime (τ) and applied electric field intensity product), and tunable compositions for desired physical/chemical properties[12,13]. Their single-crystal counterparts can be grown from low-cost solution processes with abundant raw materials. The low trap density of perovskite single-crystal devices enables a much larger sensitivity than α-Se detectors[6,12], and the corresponding µτ product is even comparable to the state-of-the-art CdZnTe single crystals[11]. However, growing a large-area perovskite single crystal from low-cost solution processes

[1]State Key Laboratory of Supramolecular Structure and Materials, College of Chemistry, Jilin University, Changchun 130012, PR China. [2]School of Mechanical and Aerospace Engineering, Jilin University, Changchun, Jilin 130022, PR China. [3]Optical Functional Theranostics Joint Laboratory of Medicine and Chemistry, The First Hospital of Jilin University, Changchun 130012, PR China. ✉e-mail: hweichem@jlu.edu.cn

for an X-ray flat panel has limitations in commercialization due to the fact of time-consuming and reproducibility concerns. To target practical X-ray imaging applications, large-area perovskite tablets, sintered wafers, and doctor-bladed polycrystal films with different compositions exhibit promising performance in a more fast and economical way, although their sensitivities are inferior to their single-crystal counterparts due to the inevitable lattice microstrain during crystallization and processing[6,12,14–21]. Furthermore, the ionic nature of the perovskite materials results in intrinsic instability of water, moisture, electric field, and other complex conditions. Accompanied by the structural instability, the polarization effect known as ions migration can seriously lead to a leakage current drift, and the optoelectronic signals are also impaired, submerging into the noise fluctuation during low dosage detection[22,23]. Low dimensional perovskites (including two-dimensional (2D), 1D, and 0D structures), whose inorganic frameworks are separated by the organic cations, are discovered to be effective in suppressing the ions migration and improving their moisture stability[24–29]. Nevertheless, the long-chain organic cations are partially detrimental to highly efficient charge carrier collection[30,31]. Besides, the water stability or long-term moisture stability of perovskite materials is still one of the biggest challenges in building multifunctional optoelectronic devices for future commercialization, which also limits their compatibility with many powerful and typical semiconductor techniques, such as photolithography, wet etching, ions implant, crystals polishing, and so on.

In this Article, we develop scalable and dimension switchable tryptamine ($C_{10}N_2H_{13}$, abbreviated as TA) lead iodide perovskite tablets with crystal structures of 1D $TA_4Pb_3I_{10}$ and 2D $TA_2PbI_4$. The 1D $TA_4Pb_3I_{10}$ perovskite is water stable due to the strong hydrophobic cations-π and π-π interactions between organic cations[32,33], which directly enables traditional semiconductor processing technologies of surface polishing, wet etching, and photolithography to fabricate sensitive X-ray flat mini-panel with electrode microelectrode arrays. The 1D/2D dimension transitions can relax the microstrain, improve the charge carrier transport with the assistance of intermolecular interactions, and realize high-performance X-ray flat panels for high-resolution imaging.

## Results

### Switchable dimension transitions and microstrain relaxation

In the TA$^+$ ions-based perovskite precursor solution, TA$^+$ ions can form different stoichiometric ratios with $[PbI_6]^{4-}$ octahedron skeletons. We discover that perovskite solution can precipitate dark yellow 1D $TA_4Pb_3I_{10}$ single crystals from the acetonitrile solvent in Fig. 1a and Supplementary Fig. 1a, if we control the crystal growth at a low crystalline rate <50 mm$^3$ min$^{-1}$. Single-crystal X-ray diffraction analysis (SXDA) confirms the 1D perovskite composition and element arrangements (See Supplementary crystallographic information file (CIF) and Supplementary Fig. 2). If we control the single crystals crystalline rate over 50 mm$^3$ min$^{-1}$, we get dark orange 2D $TA_2PbI_4$

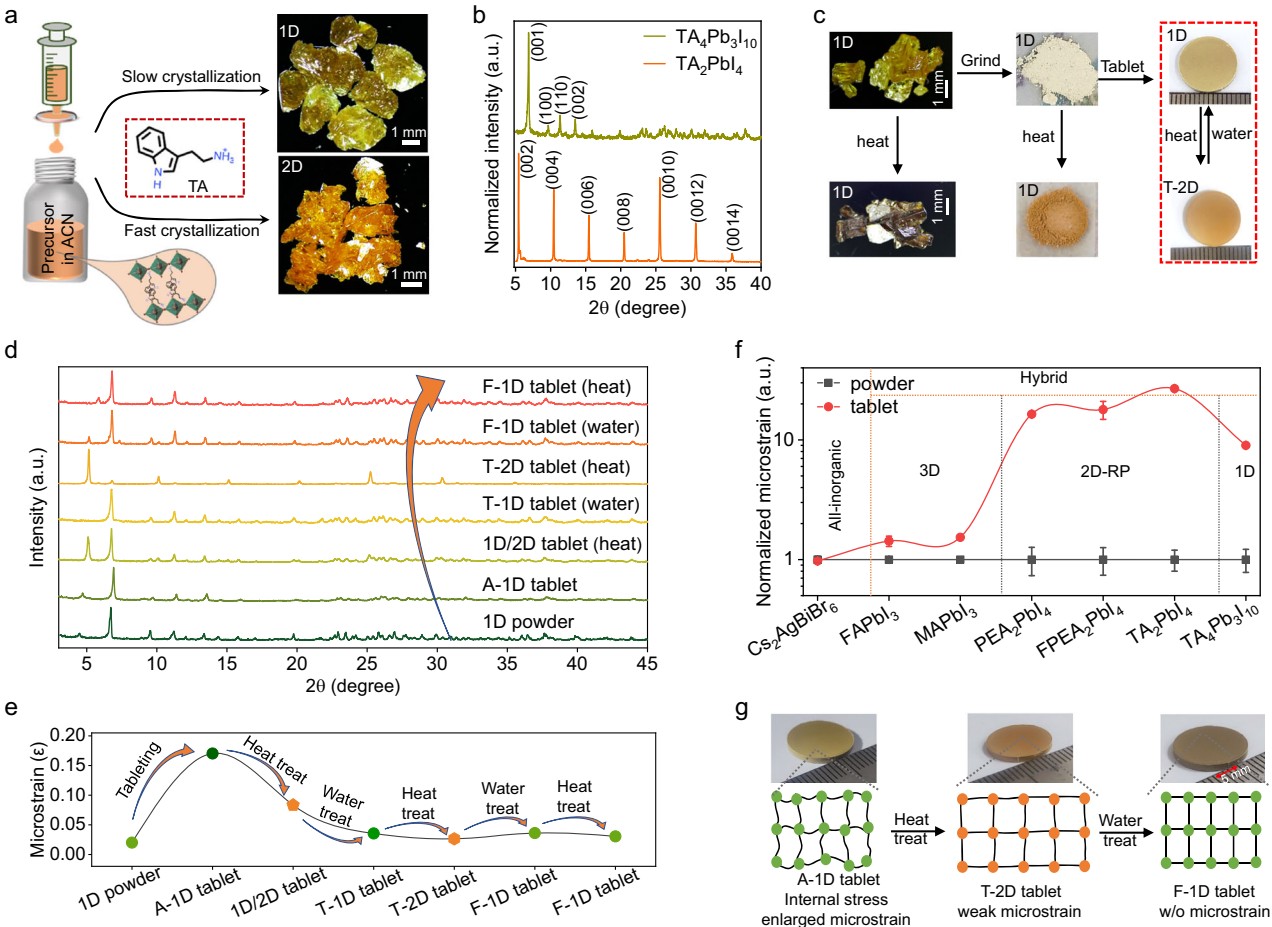

**Fig. 1 | Perovskites dimension transitions and strain relaxation. a** Scheme of crystallization of TA-based perovskites and corresponding crystals photos. **b** XRD spectra of 1D and 2D TA-based perovskites. **c** Optical photos of 1D perovskite crystals, powders, and tablets before and after heating treatment, and T-2D perovskite tablet can transform into 1D perovskite upon water soaking. **d** XRD measurements of TA-based perovskites with cycled dimension transitions. **e** Relaxation processes of the microstrain by dimension transitions. **f** Comparison of the microstrain of perovskite powder and tablet with different compositions and dimensions. **g** Schematic diagram of perovskite lattice change during microstrain relaxation processes. Inserts are corresponding photos of TA-based perovskites.

single crystals (Fig. 1a and Supplementary Fig. 1b) as evidenced by SXDA results. The corresponding X-ray Diffraction (XRD) measurements in Fig. 1b further confirm the different crystal structures. In 1D $TA_4Pb_3I_{10}$ perovskite, the $TA^+$ ions are arranged more condensed, and the ions alignments often take more time as dominated by the thermodynamic crystallization process. In contrast, the alignments of $TA^+$ ions in 2D $TA_2PbI_4$ perovskite are relatively loose, and the corresponding crystallization process should be dominated by dynamics. Since the two perovskites are composed of the same kinds of ions, we speculate that they can transform into each other at certain conditions. We grind the single crystals into powders and press the powders into scalable tablets. It is interesting to see that the as-prepared 1D (A-1D) $TA_4Pb_3I_{10}$ tablets transform to 2D-like tablets (referred to as T-2D tablets) after thermal annealing at 120 °C for 2 h (Fig. 1c and Supplementary Fig. 3), while both the 1D $TA_4Pb_3I_{10}$ single crystals and powders remain 1D structure after thermal annealing (Fig. 1c, Supplementary Figs. 3 and 4). It is worth noting that when compared with A-2D, the XRD peaks of the T-2D show obvious small-angle shift (Supplementary Fig. 5), representing a much looser packing mode exists in the T-2D tablet, and signifying much weaker intermolecular interactions between molecules. In this case, we speculate that this T-2D structure is inconsistence with A-2D, and we could get the conclusion that the dimension transition process manifested in XRD is dominated by the growth dimension of the inorganic octahedra, and the growth dimension of the inorganic octahedra is significantly affected by the assembly modes of the $TA^+$ molecules, while the assembly modes including molecular conformation and intermolecular interactions of $TA^+$ could be driven to change through the thermal energy or water treatment. It is also exciting to discover that these 2D-like tablets can transform back to 1D $TA_4Pb_3I_{10}$ tablets upon water soaking/water fumigation treatments (80 °C, RH: 99%) for ~0.5 h, which means that the 1D $TA_4Pb_3I_{10}$ perovskite tablets are water stable (Supplementary Fig. 6 and Supplementary Movie 1). Supplementary Movie 1 compares the water stability of 2D butylammonium lead iodide ($BA_2PbI_4$), phenethylammonium lead iodide ($PEA_2PbI_4$), and $TA_2PbI_4$ tablets. After soaking in water for 24 h, $BA_2PbI_4$ and $PEA_2PbI_4$ tablets quickly turn into yellow $PbI_2$ powders while the transformed 1D (F−1D) $TA_4Pb_3I_{10}$ perovskite tablet is still stable and robust. The XRD study in Fig. 1d further demonstrates the switchable dimension transitions, and the obvious intermediate mixture phase can be clearly observed in the 1D/2D tablets. After two or three cycles of dimension transitions, we find the perovskite can finally stabilize at final-state 1D (F−1D) perovskite. In addition, once the A-2D $TA_2PbI_4$ tablets are soaked in water to transform into final-stated 1D (F-1D) $TA_4Pb_3I_{10}$ tablets, the F-1D $TA_4Pb_3I_{10}$ tablets can never be switched back into 2D tablets anymore.

Since the perovskite tablets are fabricated through an external pressing process, which often induces large crystal strain and lattice microstrain, and impairs the charge carriers' transport, we then move forward to calculate the microstrain ($\varepsilon$) of the tablet samples based on the XRD characterizations by following the Williamson-Hall (W-H) calculation Eq. (1)[34,35]:

$$\text{fwhm} \cdot \cos(\theta) = \frac{K \cdot \lambda}{D} + (\varepsilon \cdot 4\sin(\theta)) \tag{1}$$

where D is the average crystallite size, $\lambda$ is the wavelength, $\theta$ is Bragg's diffraction angle. The diffraction value here should be expressed in the radian system. $\varepsilon$ is the microstrain, and K is the Scherrer constant. Microstrain is obtained from the slope of the fitting line. Fig. 1e summarizes the microstrain of the $TA^+$ ions-based perovskite samples, and the microstrain of the A-1D tablet is obviously larger than that of the 1D perovskite powders as expected. Further dimension transitions can greatly reduce the lattice microstrain, and several cycles of dimension transitions will completely relax their microstrain, serving as ideal

large-area X-ray detectors. The detailed W-H fitting lines can be found in Supplementary Fig. 7. To understand the microstrain generation and relaxation mechanism, we compared the microstrain of several typical perovskite powders and tablets, including all-inorganic cesium silver bismuth bromide ($Cs_2AgBiBr_6$) perovskite, 3D formamidinium lead iodide ($FAPbI_3$) and methylammonium lead iodide ($MAPbI_3$) perovskites, 2D $PEA_2PbI_4$, fluorophenethylammonium lead iodide ($FPEA_2PbI_4$) and $TA_2PbI_4$ perovskites, and 1D $TA_4Pb_3I_{10}$ perovskite in Fig. 1f. All the microstrains of perovskite powders are normalized to study the composition impact on the microstrains of perovskite tablets, and all-inorganic $Cs_2AgBiBr_6$ perovskite tablet shows almost unchanged microstrain, indicating the rigid lattice nature. However, small microstrain changes in the 3D perovskite after tableting as the small organic molecules are placed in the cavities of the Pb-I octahedra. In low dimensional perovskites, longer organic molecules result in obviously larger microstrain, since the layered alignments of the flexible organic molecules render them more deformation space. And this is also consistent with our previous study that organic cations adopt the main strain deformation[36,37].

Based on the above observation, the microstrain generation and relaxation processes are displayed in Fig. 1g and Supplementary Fig. 8. The 1D perovskite tablet is obtained by applying an external stress of 200−300 MPa, and corresponding internal stress causes an enlarged microstrain. As the perovskite tablets are annealed on a hotplate, external thermal energy is converted to the kinetic energy of the strained $TA^+$ ions for relaxation. During the thermal energy input process, $TA^+$ reassembled into a relatively loose packing mode, which will cause the inorganic octahedra part to fit in this new packing mode and finally self-adaptively stabilize at 2D arrangement mode, thus representing the XRD characterization of the 2D structure. In the reversible transition process, water drives the transition from 2D to the 1D structure, and we speculate that this transition process is caused by the strong hydrophobic force of cations-π interactions[38–41], and the relatively loose packing of the 2D architecture will self-assemble into the closely aligned 1D framework and further relax the microstrain. It's worth mentioning that the driving force will be transmitted once it is induced by the invasion of the water molecules, since the closely packed $TA^+$ molecules can accelerate the molecules' reassembly, and this reassembling process will not only occur on the surface for the closely packed mode, but also affect the neighboring loosely packed part to achieve a comfortable state, thus the water can drive the full phase/chemical transition. The dimension transitions between 1D and 2D tablets are essentially the energy consumption and microstrain relaxation processes upon external water/thermal stimulation.

## Perovskite dimension transitions and water stability

To further understand the perovskite dimension transition processes and study the material band edge structures, the total density of states (DOS) is performed by employing the Vienna Ab initio Simulation Package (VASP) code of density functional theory (DFT) calculation (Supplementary Data 1). The optimized differential charge density distributions in valance band maximum (VBM) and conduction band minimum (CBM) of 1D $TA_4Pb_3I_{10}$ perovskite and 2D $TA_2PbI_4$ perovskite are displayed in Fig. 2a, b, respectively. Charge carriers at the VBM and CBM of 2D $TA_2PbI_4$ are enriched in the interacting p orbitals of Pb and I atoms, indicating that the inorganic octahedrons of 2D $TA_2PbI_4$ are responsible for the charge carrier transport as expected. On the contrary, the charges at the VBM of 1D $TA_4Pb_3I_{10}$ are enriched in the delocalized p orbital of the conjugated ring in organic cations, showing the $TA^+$ can also participate in the perovskite band edge construction, and corresponding optoelectronic properties should be also affected. DFT calculations of band structures (left) and the DOS (right) of the 1D $TA_4Pb_3I_{10}$ perovskite and 2D $TA_2PbI_4$ perovskite are shown in Fig. 2c, d, respectively. The 1D $TA_4Pb_3I_{10}$ is a direct bandgap semiconductor with a bandgap of 2.8 eV, where both the VBM and CBM are located at the Z

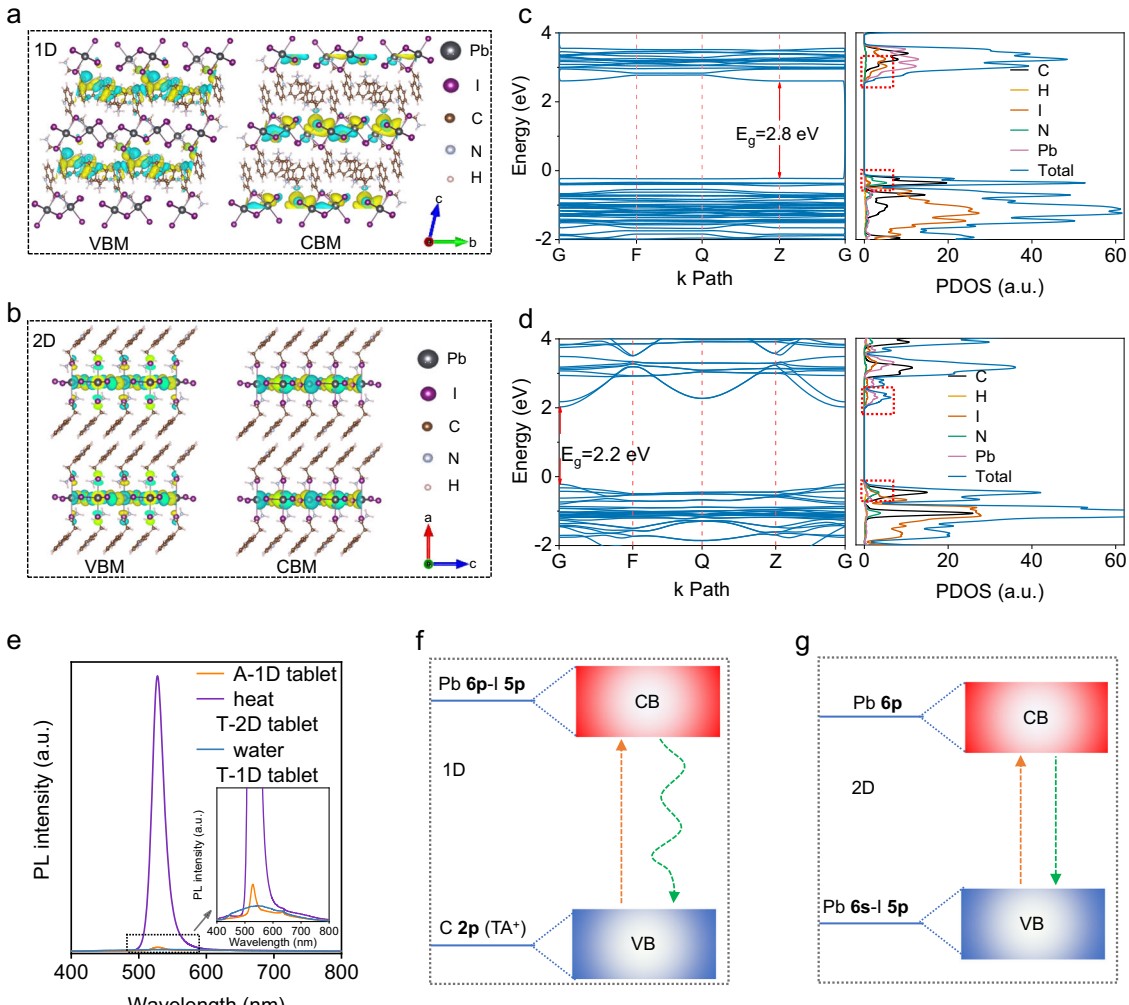

**Fig. 2 | Band edge construction of TA-based perovskites. a** The optimized differential charge density distributions of 1D TA$_4$Pb$_3$I$_{10}$, and TA$^+$ can also participate in the perovskite band edge construction. **b** The optimized differential charge density distributions of 2D TA$_2$PbI$_4$, and charges are mainly localized at inorganic frameworks. **c** Band structure and DOS simulation results of 1D TA$_4$Pb$_3$I$_{10}$. **d** Band structure and DOS simulation results of 2D TA$_2$PbI$_4$. **e** PL spectra of A-1D tablet, 1D/2D tablet and F-1D tablet. **f** Band edge construction scheme of the 1D TA$_4$Pb$_3$I$_{10}$ perovskites based on the simulation results and PL spectra. **g** Band edge construction scheme of the 2D TA$_2$PbI$_4$ perovskites.

point, and it can be clearly seen that TA$^+$ ions contribute to the valance band edge construction. However, the 2D TA$_2$PbI$_4$ shows a smaller bandgap of 2.2 eV as a direct bandgap semiconductor, and the VBM and CBM are located in the G point. Pb-I octahedrons contribute to the valance band and conduction band edge construction as shown in the DOS simulation result. The energy band structures of the TA-based perovskites are exhibited in Supplementary Fig. 9. Due to the big difference in their band edge structures, their photoluminescence (PL) varies a lot (Fig. 2e). The T-2D tablet shows a strong PL peak at 527 nm, basically consistent with the DFT calculation result. In strong contrast, the 1D TA$_4$Pb$_3$I$_{10}$ perovskite tablet exhibits an almost quenched PL emission with a broad peak, which is caused by the excimer formation and energy dissipation between closely packed TA molecules in 1D TA$_4$Pb$_3$I$_{10}$[42-44], as schemed in Fig. 2f, g. This is also consistent with the color change of the 1D TA$_4$Pb$_3$I$_{10}$ perovskite single crystals and powders in Fig. 1c, which can be explained by the fact that the stacking modes of organic TA$^+$ ions in 1D TA$_4$Pb$_3$I$_{10}$ are easily affected by the external energy, such as heat, grounding, and tableting processes, thus reconstructing the band edge structure.

The origin of the PL quenching in 1D TA$_4$Pb$_3$I$_{10}$ perovskite tablet lies in the different stacking modes and conformation alignments of the TA$^+$ ions. Supplementary Fig. 10 presents the optimized lattice

scheme of the 1D TA$_4$Pb$_3$I$_{10}$ and 2D TA$_2$PbI$_4$ perovskite according to the SXDA result, and the TA$^+$ ions align in a close-packed stacking mode due to the strong cation-π interactions and π-π interactions. Benefiting from the high electron cloud density of the pyrrole ring on the indole group, the strong cation-π interactions with distances of 3.16 Å and 3.57 Å take place between the pyrrole rings and the neighboring ethyl ammonium cations. Also, close face-to-face π-π stacking with distances of 3.73 Å to 4.67 Å occurs between the pyrrole rings and pyrrole rings, pyrrole ring and the benzene rings, benzene rings, and benzene rings. These close face-to-face stacking modes result in the typical aggregation-caused quenching effect[45]. The energy dissipation results in the PL quenching in 1D TA$_4$Pb$_3$I$_{10}$ perovskites. We further perform the PL study during the dimension transitions, and there is a broadband emission in the 1D TA$_4$Pb$_3$I$_{10}$ perovskites with low PL intensity. After thermal annealing, a strong peak gradually shows up with full width at half maxima of only ~25 nm as depicted in Fig. 3a. Insert schemes show the stacking modes change of the TA$^+$ ions during dimension transitions from 1D to 2D perovskites, the cation-π interactions totally disappear in 2D TA$_2$PbI$_4$, and π-π interactions are also largely weakened, which breaks the close face-to-face stacking modes of the TA$^+$ ions, and then greatly recover the PL emission as observed. Upon further water soaking, the 2D structure can switch back to the 1D

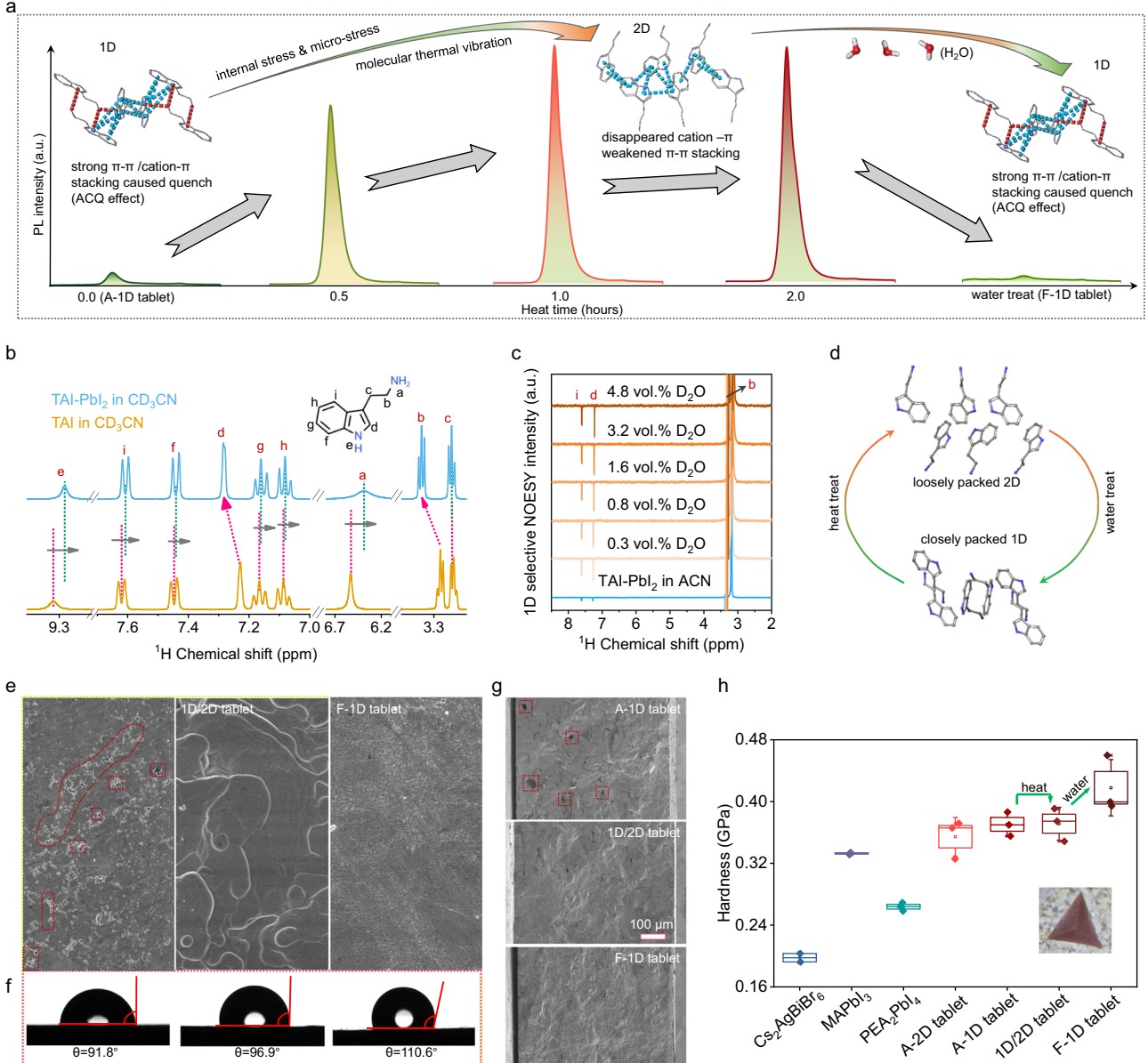

**Fig. 3 | The mechanism of the dimension transitions. a** PL transitions during the dimension transition process and corresponding molecular reassembly diagram. **b** The liquid-state 1D $^1$H NMR spectra of TAI and TAI-PbI$_2$ in deuterated ACN. **c** 1D selective NOESY spectra with titration of D$_2$O in deuterated ACN. **d** Schematic diagram of the molecular packing changes during the dimension transition process based on SXRD results. **e** Surface morphologies of A-1D tablet, 1D/2D tablet and F-1D tablet. After cycles of dimension transitions, the F-1D tablet with no microstrain and pinholes is fabricated. **f** Contact angles characterization of TA-based perovskite tablets. F-1D tablet exhibits a larger water contact angle of 110.6°. **g** Cross-section SEM images of the TA-based perovskite tablets. The F-1D perovskite tablet represents a compact bulk with no pinholes, while the A-1D perovskite tablet shows many irregular holes. **h** Hardness of classical 3D MAPbI$_3$, 2D PEA$_2$PbI$_4$, all-inorganic Cs$_2$AgBiBr$_6$, and TA-based perovskite tablets.

structure, accompanied by almost quenched PL again, representing the recovery of the close packing modes.

We perform Nuclear Magnetic Resonance Spectroscopy (NMR) measurements to confirm the intermolecular interactions. Two-dimensional (2D) $^{13}$C{$^1$H} heteronuclear correlation (HETCOR) experiments use through-space dipolar couplings to correlate the isotropic chemical shifts of nearby (<1 nm) $^1$H and $^{13}$C nuclei. As mentioned above, after thermal annealing, the A-1D structure transforms into the T-2D structure, accompanied by the disappearance of cation-π interactions and weakened π-π interactions. To confirm this, 2D $^{13}$C{$^1$H} HETCOR experiments are performed for the bulk A-1D perovskite (before thermal annealing) and bulk T-2D perovskite (after thermal annealing). As depicted in Supplementary Fig. 11, although the solid-state $^1$H spectra of both structures do not distinguish peaks of hydrogen atoms in different chemical environments, we can clearly see that the chemical shift (δ) of C1 in the $^{13}$C spectrum of the T-2D shifts to a higher field, representing the increase of the shielding constant in the T-2D case, and we attribute this shielding constant increase to the demagnetization anisotropy for the de interaction of C1 and adjacent pyrrole/indole ring, which means the disappearance/weakness of cation-π interactions. The disappearance of the cation-π interactions after thermal annealing is also confirmed by the IR measurements (Supplementary Fig. 12), since the infrared vibration peak of H-N-H bonds shows up, and disappears again upon water soaking treatment.

In order to further demonstrate our speculation that the transition from 2D perovskite to 1D perovskite upon water soaking is driven

by strong hydrophobic interaction of cation-π interactions. We perform the 1D selective NOESY measurements with titration of $D_2O$ in deuterated acetonitrile (ACN). We first check the NMR spectra of TAI and TAI-PbI_2 in deuterated ACN and deuterated dimethyl sulfoxide (DMSO). There is no δ change between pure TAI and TAI-PbI_2 samples in DMSO (Supplementary Fig. 13). While the δ of the TAI-PbI_2 sample shifted compared to pure TAI molecules in ACN solvent (Fig. 3b), representing the existence of intermolecular interactions between TAI and PbI_2 even in the ACN solution. This transition state of the TAI-PbI_2-based perovskite is suitable for monitoring stimulus-response, so we use this deuterated ACN system to perform the 1D selective NOESY experiment for $D_2O$ titration experiments. As shown in Fig. 3c, when we excite hydrogen b (hydrogen on carbon connected to primary amine), the chemical shift peaks of hydrogens d and i appear, which means that hydrogens d and i are spatially correlated to hydrogen b (blue line in Fig. 3c). With titration of $D_2O$ into the ACN solution, the correlation intensity gradually increases and tends to be stable, demonstrating that water drives the molecule's alignment more closely as shown in the schematic diagram of Fig. 3d. In this process, water doesn't enter the crystal lattice according to the IR study in Supplementary Fig. 12.

To study the morphology change of perovskite tablets during the dimension transition, we perform the Scanning Electron Microscope (SEM) and Atomic Force Microscope (AFM) in Fig. 3e and Supplementary Fig. 14, respectively. Obvious pin holes can be observed on the surface topography of the A-1D perovskite tablet. After thermal annealing, the A-1D perovskite tablet transforms into mixed 1D/2D perovskite with layered structures showing up on the tablet surface. Upon dimension transition from mixed 1D/2D perovskite to T-1D or F-1D perovskite, a hole-free and smooth surface appear. The corresponding contact angles of water on the tablet surfaces on A-1D, 1D/2D, and F-1D perovskite tablets are displayed in Fig. 3f, respectively, and the F-1D perovskite tablet exhibits the largest contact angle of 110.6°, further confirming the hydrophobic interaction in 1D $TA_4Pb_3I_{10}$ perovskite. We also check the cross-section SEM images of these three samples in Fig. 3g, consistent with the surface topography result, small holes can be observed in A-1D tablets, while the T-1D tablet is compact with no pin holes.

Based on the perovskite dimension transition process and dimension-dependent interaction difference, we speculate a more thermally and mechanically stable F-1D $TA_4Pb_3I_{10}$ perovskite tablet compared to the A-1D $TA_4Pb_3I_{10}$ or 2D $TA_2PbI_4$ perovskite tablets. Thermogravimetric analysis (TGA) measurements of TAI, PbI_2, A-1D, T-2D, and F-1D tablets are performed under an elevating temperature rate of 5 °C min⁻¹ under N_2 flow. As exhibited in Supplementary Fig. 15, the initial decomposition temperatures of all the TA-based perovskite tablets are higher than 200 °C, while the TAI tablet is only 131 °C, indicating the large bonding strength of TAI and PbI_2. In addition, the decomposition temperature for the F-1D perovskite tablet is 242 °C, as expected, the highest for the TA-based perovskite tablets. Except for the good thermal stability, we also observed better mechanical properties in TA-based perovskite tablets. Nanoindentation measurements are carried out to evaluate the hardness of the TA-based perovskite tablets. Fig. 3h summarizes the hardness of the various perovskite tablets. TA-based perovskites possess larger hardness than classical 3D MAPbI_3, 2D PEA_2PbI_4, and all-inorganic $Cs_2AgBiBr_6$ perovskites, further confirming the stronger bonding exists in TA-based perovskites. Supplementary Fig. 16 shows the indentation curves of the tablets, and we can clearly see that all the indentation depths of MAPbI_3, PEA_2PbI_4, and $Cs_2AgBiBr_6$ perovskites are deeper than 3000 nm. In contrast, TA-based perovskite tablets show shallower than 3000 nm with stronger resistance to plastic deformation. Besides, we analyzed the anisotropic atom displacement parameters ($\bar{U}_{eq}$) of these classical perovskites in Supplementary Table 1, where $TA_4Pb_3I_{10}$ shows a much smaller $\bar{U}_{eq}$ than MAPbI_3, PEA_2PbI_4, or all-

inorganic $Cs_2AgBiBr_6$ perovskites, further indicating the strong bonding condition in TA-based perovskite.

## Water-stable and high-performance X-ray detectors

The F-1D tablet with good crystallinity is very stable in water, and Fig. 4a shows the water-driven dimension transition processes from T-2D perovskite to F-1D perovskite. After soaking in water within 2 h, the T-2D perovskite tablet can completely transform into F-1D perovskite tablet as evidenced by the XRD results in Fig. 4a (right panel), and corresponding photos of perovskite tablets are shown in the left panel of Fig. 4a. To confirm the excellent water stability of the F-1D perovskite tablets, we soak the tablets in water for different time, followed by thermal annealing at 120 °C for ~3 h to completely dry the perovskite tablets, and then fabricated devices with the architecture of Cr/BCP/C_{60}/perovskite/Au to suppress the dark leakage current and extract the photogenerated carriers. Fig. 4b shows the sensitivity of F-1D perovskite tablets with different water soaking time. The sensitivity of the F-1D tablet increases as the water soaking time, stabilizes at $1.1 \times 10^5 \mu C \ Gy_{air}^{-1} \ cm^{-2}$ after water soaking for 48 h, and gradually decreases to $5.7 \times 10^4 \mu C \ Gy_{air}^{-1} \ cm^{-2}$ after 720 h of water soaking (Supplementary Fig. 17). The F-1D perovskite tablets can be stable in water at least for over 40 days before decomposition into PbI_2 (Supplementary Fig. 18). For a flat panel detector, noise equivalent dose (NED) is more relevant for a medical application since the NED takes the integration time into account, representing the minimization dose which is significantly important to the medical imaging patient[46–48]. We confirm the NED of our X-ray flat panel detector based on the reported noise experiments methods (Supplementary Fig. 19)[46], The NED of the F-1D detector can reach 118 nGy, comparable with the reported scintillator detectors 4336WXv2, 4336Rv3, 4336Wv4 of 459 nGy, 666 nGy, and 566 nGy[48]. Besides, we also focus on the detection limitation here. The signal-to-noise ratios (SNRs)[49] is one of the most important metrics of a detector to evaluate its detection limitation of dose rate, the SNRs of the F-1D tablets under various dose rates are displayed in Fig. 4c, which are also relatively stable for the different soaking time in the water, demonstrating the excellent water stability of the F-1D tablet to the best of our knowledge. In addition, the F-1D tablet is also operated from room temperature (23 °C) to as high as 150 °C to check the operation stability under various working temperatures. As exhibited in Fig. 4d and Supplementary Fig. 20, with the working temperature increasing, the sensitivity of the A-1D tablet decreases from $4.3 \times 10^4$ to $3.3 \times 10^4 \mu C \ Gy_{air}^{-1} \ cm^{-2}$. While the sensitivity of the F-1D tablet increases from $7.3 \times 10^4$ to $9.6 \times 10^4 \mu C \ Gy_{air}^{-1} \ cm^{-2}$ (Supplementary Fig. 21), better thermal stability of F-1D demonstrates a more rigid structure of the F-1D tablet. A similar phenomenon is also observed in the SNRs of the A-1D tablets and F-1D tablets at elevated temperatures in Fig. 4e, and the reduced SNRs for A-1D devices should be related to the residue microstrains, highlighting the more integrity lattice and stronger bonding strength of the microstrain-free F-1D tablet. Furthermore, the μτ product of the F-1D tablet is $1.8 \times 10^{-4} \ cm^2 \ V^{-1}$ at room temperature (Supplementary Fig. 22), a relatively small value, but the μτ product only tells how fast the charge carriers can diffuse in the material. The robust crystal structure of the F-1D tablet enables negligible ions migration phenomenon even under a large applied bias of 1000 V, which corresponds to a large electric field E of 800 V mm⁻¹, contributing to a large μτE product (or schubweg distance) of 1.44 cm. The μτE product, the mean drift length of the charge carrier, is actually the most important Figure-of-the-Merit to determine the charge collection efficiency/sensitivity[50,51], this is why the amorphous-Se with a small μτ product can be commercialized. The large μτE of 1.44 cm in F-1D tablet devices under a high applied bias of 1000 V enables the implementation of high sensitivity in F-1D devices. Besides, the μτE of 1.44 cm is already limited by the maximized output of our source power at 1000 V. Further increase of the device sensitivity could be promised with larger applied bias voltage, as shown in

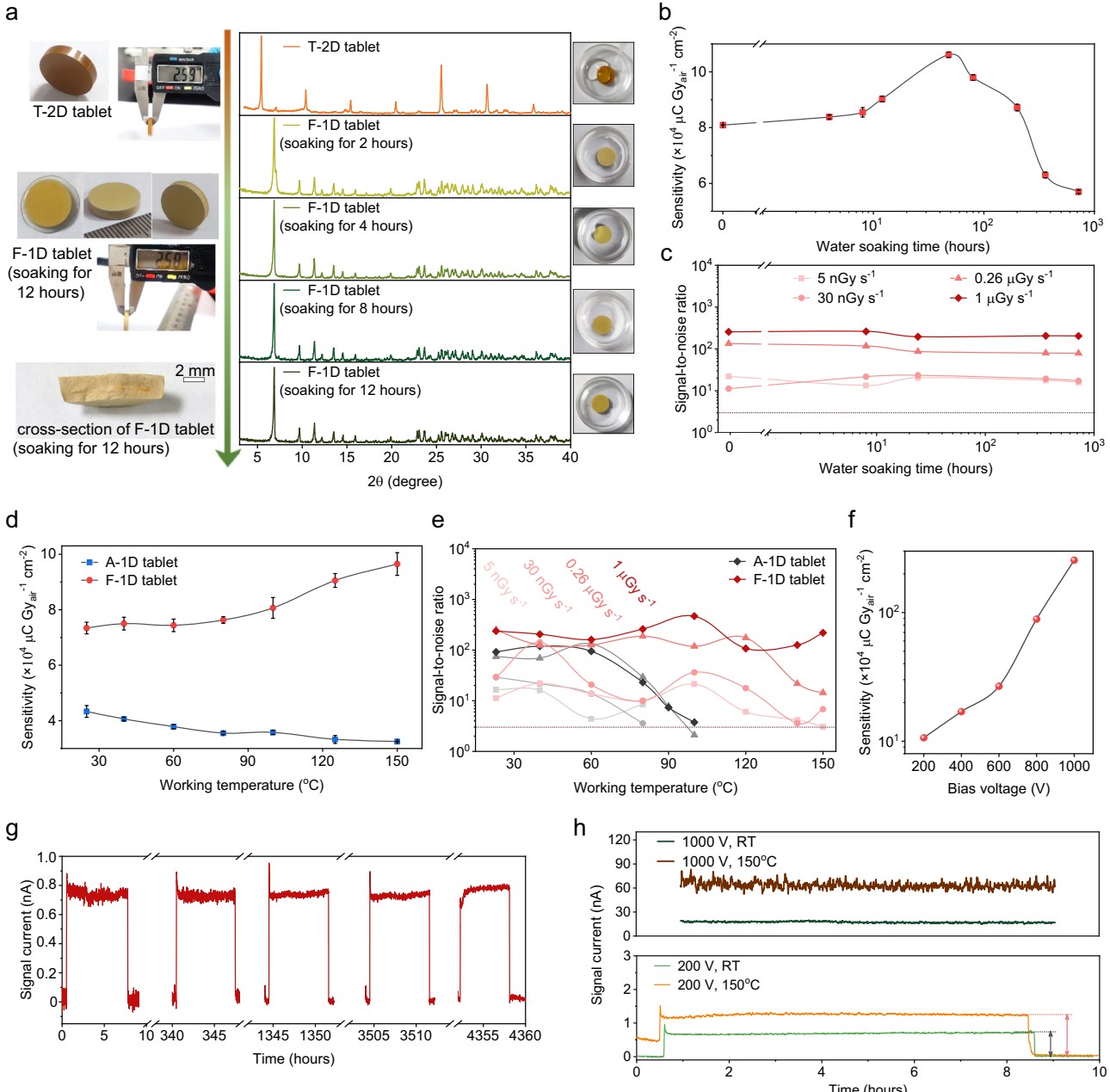

**Fig. 4 | Water and thermal stability of TA-based perovskite tablets. a** Water-stable and heat-stable characterization by XRD. **b** Water-stable sensitivity of F-1D tablet. **c** SNR characterizations of F-1D tablets treated with water under various dose rate conditions. **d** Heat-stable sensitivity of F-1D tablet. **e** SNR characterizations of F-1D tablets at different working temperatures under various dose rate conditions. **f** Sensitivity curve with bias voltage. **g** Long-term air stability of F-1D tablet (160 V mm$^{-1}$, 1.08 μGy$_{air}$ s$^{-1}$). **h** Operational stability of F-1D tablet under room temperature (RT) and 150 °C, when applied with the electrical field intensity of 160 V mm$^{-1}$ and 800 V mm$^{-1}$ (1.08 μGy$_{air}$ s$^{-1}$). All the device thickness is ~1.25 mm, all the device area is 4 mm$^2$.

Fig. 4f, Supplementary Figs. 23 and 26. The highest sensitivity of the X-ray detector is confirmed to be $2.5 \times 10^6$ μC Gy$_{air}^{-1}$ cm$^{-2}$. A photo-conductive gain of ~500 exists in the X-ray detector under the applied external bias voltage of 1000 V, calculated as below:

$$\text{Theoretical sensitivity} = \frac{e}{W_{\pm} \times \left(\frac{\mu_{en}}{\rho}\right)} \qquad (2)$$

$$W_{\pm} = 2.8 \times \text{bandgap} + 0.5 \qquad (3)$$

Where e is the elementary charge, $\frac{\mu_{en}}{\rho}$ is the mass energy-absorption coefficient in the air of 120 keV X-ray, obtained from the NIST database,

$W_{\pm}$ is the amount of radiation energy consumed per electron-hole pair generated in a semiconductor[52]. Bandgap refers to the bandgap of the response material, in the 1D TA$_4$Pb$_3$I$_{10}$ case, it is 2.8 eV. In addition, the trapped carrier type in this photoconductive gain device is assessed by space charge-limited current (SCLC) measurements, and the result indicates that the trapped type carrier is the electron, and the transport carriers are holes (Supplementary Fig. 24). the response time of 172 μs ($\tau_{on}$) and 128 μs ($\tau_{off}$) is also characterized under the applied electric field intensity of 800 V mm$^{-1}$ (Supplementary Fig. 25). Though high gain exists in the F-1D tablet detector, a large charge carrier mobility and electric field product can still result in a small transit time, thus an acceptable response time (charge carrier lifetime) is achieved in the F-1D detector (Supplementary Fig. 25). Furthermore,

Considering the anisotropic property of this TA-based low-dimensional perovskite material, we supply the anisotropic characterization of $TA_4Pb_3I_{10}$ single crystal and F-1D tablet in Supplementary Fig. 26. Also, we conclude the sensitivity variation with applied electric field intensity of the 1D single crystal and F-1D tablet in Supplementary Fig. 26e, which represents that there is a little difference between the sensitivity of three carrier transport planes in 1D single crystal. This should result from the band edge structure construction of 1D $TA_4Pb_3I_{10}$ perovskite. Besides, the sensitivity of the tablet detector will be comparable to that of the single crystal when the applied electric field intensity is large enough to exceed 660 V mm$^{-1}$, which means that a sufficiently large $\mu\tau E$ product can achieve a result where the $\mu\tau$ could be ignored.

Finally, the long-term air stability of the F-1D detector is also inspected. As exhibited in Fig. 4g, the signal response of the F-1D tablet device has almost no degradation during about six months of inspection under ambient air conditions without any encapsulation, indicating excellent device air stability. Besides, the operation stability under working conditions (200 V, RT) and extreme conditions (1000 V, 150 °C; 1000 V, RT; 200 V, 150 °C) are also performed. The F-1D tablet X-ray detector shows no ion migration or signal degradation during the working period (Fig. 4h), demonstrating their super operation stability under high-temperature/electric field conditions. The excellent stability can be assigned to the robust crystal alignments and the microstrain-free structure of the F-1D tablet.

## Water-stable perovskite X-ray flat panel and X-ray imaging

Encouraged by the excellent water stability and sensitivity of the perovskite tablets, we fabricated a perovskite X-ray flat mini-panel with uniform and compact F-1D perovskite tablets. The excellent water stability of the F-1D perovskite tablet enables us to adopt many traditional and powerful semiconductor technologies to fabricate high-performance X-ray flat panels. We first polished the F-1D perovskite tablet with water for further electrode deposition. Considering the excellent water stability and acid-base stability, we adopt traditional ultraviolet lithography technology to construct microelectrode arrays for an X-ray panel imager with a diameter of 1.3 cm/2.4 cm × 3.6 cm (Fig. 5a, b), and the pixel size is around 100 μm × 100 μm square. Fig. 5b displays the flowing steps by water during the photolithography process, and we have no limitation to scale up the microelectrode arrays X-ray panel imager to a large area of 3.6 cm × 2.4 cm (Fig. 5b and Supplementary Movie 2). Large areas of X-ray flat panels can be easily realized by employing industrial tablet presses. In our imaging system, we design a probe card as the signal readout terminal as shown in Supplementary Fig. 27, and 100 probes with a needle arm of $\phi = 200$ μm, and a needle tip of $\phi = 50$ μm are aligned together with 300 μm spacing distance to match the electrode arrays of 200 μm pixel size with 100 μm spacing distance for large enough signal from a read-out circuit. Fig. 5c, d, e display circuit photos, the enlarged probes, and the signal transmission circuit diagram, respectively, and X-ray imaging by linear detector arrays can be realized by following one-direction linear scanning. Fig. 5f shows the optical photographs of a tooth as a target for X-ray imaging, and clear dental X-ray imaging of the tooth is shown in Fig. 5g. Further improvement can be expected if 2D detector arrays are employed, and the applied X-ray doses by 2D detector arrays are around 10−20 μGy.

To further approach the detection limit of the X-ray panel imager, we shortened the pixel spacing to 20 μm, and collected the signal charges in the lateral direction to image the standard X-ray lead bar phantoms in Fig. 5h, and the corresponding X-ray image is clearly shown in Fig. 5i. Figure 5j is the enlarged area to show the spatial resolution in the range of 7.0−10.0 lp mm$^{-1}$, and the lead line pairs in 10.0 graduation can be clearly distinguished. Since the upper limit of the standard X-ray lead bar phantoms is 10.0 lp mm$^{-1}$, we calculate the

device spatial resolution from the X-ray image to fairly evaluate X-ray imaging capability, and we realize a record spatial resolution of 17.2 lp mm$^{-1}$ for perovskite X-ray detectors, to the best of our knowledge. Although ignoring the current cross-talk between pixels, this single-pixel scanning result represents the comprehensive performance of a detector. Actually, the high resolution needs to be realized with a tiny device pixel size, but a small device size/area definitely reduces the signal amplitude. So high device sensitivity is a requisite for high-resolution imaging, and the high resolution here further confirms the device sensitivity. Stable and high device sensitivity must be required to meet the minimum signal value that the imaging system can stably capture and to distinguish the weak dose rate changes of the X-ray obscured by the image object.

In conclusion, we develop a water-stable halide perovskite material with two kinds of switchable structures. The microstrain of scalable 1D $TA_4Pb_3I_{10}$ perovskite tablets can be relaxed by dimensional transition processes, which endowed the tablet with a large tolerance for high electric field, resulting in a comparable $\mu\tau E$ with high-quality single crystals, thus, enabling a large sensitivity of $2.5 \times 10^6$ μC Gy$_{air}^{-1}$ cm$^{-2}$, a lowest detectable dose rate of 5 nGy$_{air}$ s$^{-1}$. The prominent air/thermal/electric field stability of the perovskite panel imager can compete with commercial scintillators due to their superior resistance to ion drift even at a high temperature of 150 °C and high electric field intensity of 800 V mm$^{-1}$. Benefiting from the excellent water stability of 1D $TA_4Pb_3I_{10}$ perovskite, we realize an X-ray flat mini-panel imager with excellent X-ray imaging spatial resolution of 17.2 lp mm$^{-1}$. The low-cost and scalable perovskite panels are more practical for industrial production in future commercialization.

## Methods

Materials and synthesis of TAI and TA-based perovskite can be seed from Supplementary Methods. Spectra characterization ($^1$H NMR, $^{13}$C NMR, and MALDI-TOF mass) of molecule TAI is shown in Supplementary Figs. 28−30.

### Extrusion molding of the perovskite tablets and device structure of the X-ray detector

The pressure applied to the powder materials is 200−300 MPa, and the extruded inner cavity is a cycle with a diameter of 13 mm or a rectangle of 36 mm × 24 mm. The device structure of the X-ray detector is Au (50 nm)/perovskite/$C_{60}$ (25 nm)/BCP (8 nm)/Cr (30 nm). $C_{60}$/BCP/Cr/Au was subsequently evaporated. The photoconductive device with gain is often accompanied by a large dark current. To suppress the dark current, we designed the $C_{60}$/BCP thin layer between perovskite and electrode Cr as the blocking layer of the injection of external electrons under dark conditions. The perovskite layer will generate free charges upon X-ray irradiation, and part of the generated electrons will be trapped by shallow defects to induce a photoconductive gain, which results in a high device sensitivity.

### Characterization

XRD measurements were performed by Rigaku X-ray diffractometer (SmartLab (3)). SEM measurements were confirmed by Regulus8100 (Hitachi). Water contact angles were measured in air by Dataphysics OCA20. TGA measurements were checked by Q500-TA. Hardness was measured by a nanoindentation instrument (DUH-211, SHIMADZU, Japan) with a pyramidal indenter (pyramidal indenter, TOKYO DIAMOND Tools MFG. Co., Ltd., Japan). AFM measurements were supported by the instrument BRUKER ICON-XR, and measured with RTESPA-300 type tips. High-Resolution Ultraviolet Photoemission Spectra (UPS) were measured by an integrated ultrahigh vacuum system equipped with a multi-technique surface analysis system (VG ESCALAB MK II spectrometer). Infrared Spectroscopy (IR) was measured by VERTEX 80 V-ATR. The photolithography and oxygen

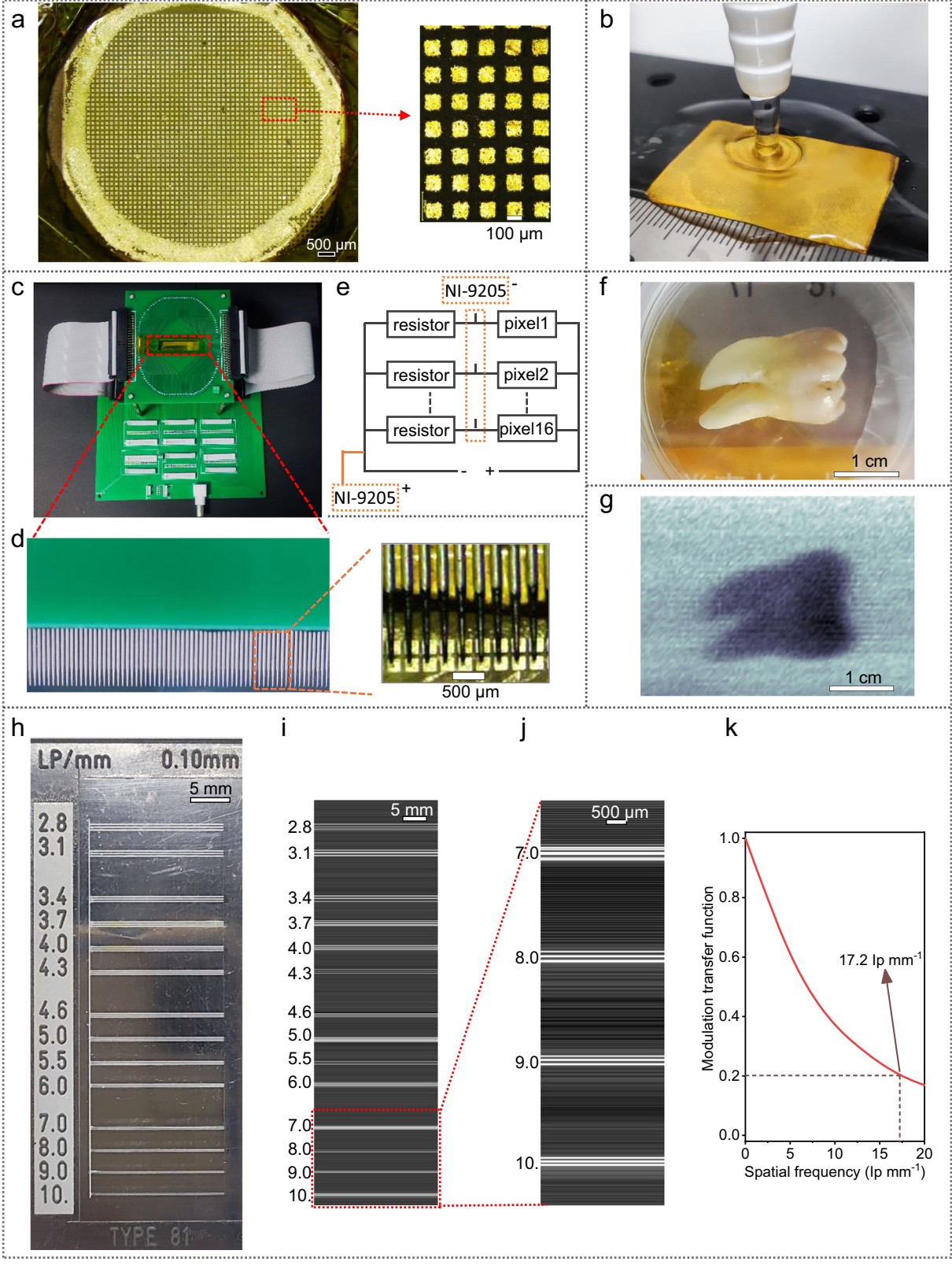

reactive ion etching (RIE) process was achieved by Plasmalab Oxford 80 Plus system.

## X-ray characterization
The X-ray tube with a tungsten target is MGI 320 instrument made by YXLON. The X-ray focal size (the diameter of the metal target) is controllable from 0.4 to 1 mm, all X-ray experiments in this work were tested under the setting of focal size of 1 mm. Under this focal size condition, the corresponding diameter of the test area at a linear distance of 500 mm from the focal spot is 360 mm. The dose rate of the continuous X-ray can be gradually tuned by tube voltage and tube current, and the tube voltage of the X-ray source is fixed at 120 kV$_p$, dose rate of the X-ray is tuned by changing the tube current (from 0.01 mA to 15 mA). A Pb plate with a thickness of ~6 mm is applied to

**Fig. 5 | Photolithographic microelectrodes and X-ray imaging. a** Scalable X-ray panel imager with microelectrode arrays fabricated by the photolithography process. **b** One of the flowing water scouring steps during the photolithography process. **c** A photo of the imaging system for the X-ray panel imager. **d** The enlarged photos of the probe arrays (left) and probe arrays in contact with the electrode surface (right). **e** The circuit design of the readout probe card. Each pixel is connected with a resistor of 20 GΩ in series connection mode, and a signal acquisition card of NI-9205 with several channels serves as a terminal to read out the signal voltage of every resistor connected to its different channels. **f** Optical photograph of a tooth. **g** Dental X-ray imaging examination. Imaging conditions: the tube voltage is 70 $kV_p$, the electric field intensity applied to the device is 800 V $mm^{-1}$, and the device area is 0.2 × 0.2 $mm^2$. **h** Optical photograph of a standard X-ray lead bar phantoms. Imaging conditions: the tube voltage is 120 $kV_p$, the electric field intensity applied to the device is 10 V $\mu m^{-1}$, the device area is 1.5 × 1.25 $mm^2$. **i** An X-ray image of the standard X-ray lead bar phantoms. **j** Enlarged X-ray image to show the clear spatial resolution range of 7.0–10.0 lp $mm^{-1}$. **k** The modulation transfer function for the X-ray detector based on edge spread function measurement of the X-ray image. The thickness of all the above tablets/devices is ~1.25 mm.

further attenuate the X-ray dose rate to target low-dosage imaging applications. Before we perform the X-ray characterization, we carefully calibrated the dose rate of the X-ray source at every tube current with IBA MagicMaX Multi-Detector (XR 40–150 kV). A Keithley 2400 was used to apply a bias voltage (0–200 V). A DC power KXN-10002D (ZHAOXIN) was used to apply the bias voltage between 200 V–1000 V, and the low-noise current preamplifier (SR570, Standford Research Systems) and SR830 lock-in amplifier were used to record the signals. The standard X-ray lead bar phantoms are imaged by a single pixel scanning assisted with 2D displacement platform. The tooth is imaged by the NI-9205 imaging system.

### NMR measurements
Solid-state NMR characterization was performed with a Bruker AVANCE III 600 MHz NMR spectrometer. CP-MAS $^{13}$C NMR spectra were recorded under a 10 kHz spinning rate. To prepare the T-2D powder samples, we first tableted the A-1D perovskite tablet, and then thermally annealed it at 120 °C for ~4 h to get a T-2D perovskite tablet. And finally ground the T-2D tablet into T-2D powder for NMR measurements. Liquid NMR measurements were characterized by Bruker AVANCE III 600 MHz NMR spectrometer and QONE AS 400.

### Photolithography
70 nm Au was deposited on the F-1D tablet, then positive photoresist (aqueous solution) was spun onto the tablet at 3000 r.p.m for 30 s, followed by heating at 88 °C for 30 min under dark conditions. The pattern on the mask was transferred to the photoresist by ultraviolet (365 nm, 350 W) exposure for 12 s. Development was carried out by immersing the tablet in developing solution (aq) for 8 s to remove the exposed photoresist, followed by rinsing with deionized water and drying with $N_2$ flow.

### Wet RIE
The irradiated tablet was immersed in 0.5 wt% NaOH (aq) for 8 s to remove the exposed photoresist, and then cleaned the NaOH (aq) with flowing water for around 30 s and dried the tablet with $N_2$ flow.

### Gold etching and unexposed photoresist peeling
The above-dried tablet was immersed in the gold etching solution for 20 s, and cleaned the gold etching solution with flowing water for 30 s, and dried the tablet with $N_2$ flow. Finally, the tablet was immersed in EtOH for 1–2 min to peel off the unexposed photoresist.

### Density functional theory simulations
The first-principles calculations were carried out via the VASP. The electron-ion interaction was described using projected augmented-wave pseudopotentials. The generalized gradient approximation formulated by Perdew-Burke-Ernzerhof was performed as the exchange-correlation function. The atomic positions were fully relaxed until the force on each atom was smaller than 0.01 eV and the convergence threshold for the self-consistent field was $10^{-4}$ eV. The Fermi level of each system was set at the maximum position of the valence band to perform the calculation of the DOS. Molecular distances were displayed with Schrödinger-Maestro.

## Data availability
The data that support the plots within this paper are available from the corresponding author upon request. Source data are provided with this paper. The X-ray crystallographic coordinates for structures reported in this study have been deposited at the Cambridge Crystallographic Data Centre (CCDC), under deposition numbers CCDC 2224700. These data can be obtained free of charge from The Cambridge Crystallographic Data Centre via www.ccdc.cam.ac.uk/data_request/cif. Crystallographic data for the structures reported in this Article have been deposited at the Cambridge Crystallographic Data Centre, under deposition numbers CCDC 2224700. These data can be obtained free of charge from The Cambridge Crystallographic Data Centre via www.ccdc.cam.ac.uk/data_request/cif. Source data are provided with this paper.

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

## Acknowledgements

We thank Prof. Daowei Li at Norman Bethune Stomatological School of Jilin University for providing the tooth for X-ray imaging. This work is financially supported by the National Natural Science Foundation of China (22105083 and 52173166), the Project of Science and Technology Development Plan of Jilin Province (20230101025JC), and the Fundamental Research Funds for the Central Universities, JLU and JLUSTIRT (2017TD-06).

## Author contributions

H.W. conceived and supervised the project. W.P. performed most of the experiment during the whole process. Y.H. assisted with the imaging system. W.L. performed the DFT simulation, and L.L. helped with the transition mechanism analysis. K.G. and B.L. carried out the single crystal analysis. J.L.Z. and J.H.Z assisted in the photolithography experiments. C.W. and H.H. carried out the nanoindentation test. B.Y. commented the results and provided constructive suggestions. H.W. and W.P. wrote the manuscript, and all the authors commented and reviewed the manuscript.

## Competing interests

The authors declare no competing interests.
