## [Peer Review File · Nature Communications]

Cations- π Interactions Enabled Water-stable Perovskite X-ray Flat Mini-panel ImagerReviewers' comments:

Reviewer #1 (Remarks to the Author):

my review is attached as Word file below

Reviewer #2 (Remarks to the Author):

The authors show transition between (1D) tryptamine lead iodine (TA4Pb3I10) and 2D TA2PbI4 perovskite tablets. Such transition was made possible because of interplay of water/moisture, cation- π and π - π transitions. The discussed chemistry, and how it influence the electronic band structure is interesting. Subsequently, they show how the water-stable perovskites can be used for X-ray detection.

The work is compelling and can be published in Nature Communication. However, a few clarifications are required before that.

1. Clarification on structure.

(a) What exactly is the crystal structure for the 1D case? Does it qualify to be called as perovskite structure?

(b) After water soaking experiments (water-stable perovskites), does the structure contain stoichiometric amount of water in the crystal? How does water is incorporated in the sample? Also refer to very relevant paper: "Releasable Water Charge-Trapping and Water-Resistant Photodetection Using 1d Perovskitoid Hydrate Single Crystal. Adv. Mater. 2022, 34, 2204710."

(c) Authors are suggested to carry out TGA and DSC measurements.

(d) Can author provide the check-cif files in supporting information? What is the status of A-, B-, C-alerts?

2. Clarification on charge transport.

(a) Do authors suggest that the hole transport through the organic part and electron transport through the inorganic part? Refer to the very relevant paper "Combining π -Conjugation and Cation- π Interaction for WaterStable and Photoconductive One-Dimensional Hybrid Lead Bromide J. Phys. Chem. Lett. 2023, 14, 1870-1876".

(b) Do the water molecules influence charge transport? Refer: "Adv. Mater. 2022, 34, 2204710."

(c) I would expect poor carrier mobility from the present hybrid 1D structure, compared to 3D hybrid Pb-I based perovskites, reducing the X-ray detection efficiency of the 1D structure. Any comment?

3. To me, the larger part of the manuscript is based on controlling/switching the structure-property relationship, by controlling various intermolecular interactions. The title of the manuscript does not reflect that part. The title highlights the device part, which is important but probably a smaller contribution in the manuscript.

4. There are other similar approaches to improve the water stability of hybrid perovskites. Refer to: Adv. Energy Mater. 2015, 5, 1501066; ACS Omega 2020, 5, 46, 29631; J. Am. Chem. Soc. 2021, 143, 19901; and so on. Authors might mention those approaches in the revised manuscript, comparing the efficacy of their present approach.

Reviewer #3 (Remarks to the Author):

In this work, the authors reported a water-stable one-dimensional (1D) tryptamine lead iodine (TA4Pb3I10) perovskite tablet, which can be switched to 2D TA2PbI4 perovskite tablet through thermal annealing or water-soaking treatments. This is an interesting discovery, however the understanding of mechanism of this phenomenon is insufficient to understand the device performance. The reported ut product of the best tablet in this work is $1.8 \times 10^{-4} \text{ cm}^{-2} \text{ V}^{-1}$ is not very good, and not significantly better than other best reported single crystals, such as MAPbI3, MAPbBr3, FAPbBr3, and FAPbI3 etc. It is not clear to this reviewer why these tablets would have much higher sensitivity of $2.5 \times 10^6 \mu\text{C Gyair}^{-1} \text{ cm}^{-2}$, low detection limit of 5 nGyair s^{-1} and high spatial resolution of 17.2 lp/mm, or it is just because inappropriate characterizations/claims. Considering the above points and the following concerns, I don't recommend publishing this work in this journal. Below are some detailed comments. The concerns/questions are listed below:

1. The authors reported a high sensitivity of $2.5 \times 10^6 \mu\text{C Gyair}^{-1} \text{ cm}^{-2}$. Similar X-ray sensitivity have already been reported by other methods or compositions. For example, Shen et al. reported a sensitivity of $(2.5 \pm 0.2) \times 10^6 \mu\text{C Gyair}^{-1} \text{ cm}^{-2}$ (Shen et al., Nat. Photonics 2022, 16, 575), Dong et al. reported a higher sensitivity of $5.2 \times 10^6 \mu\text{C Gyair}^{-1} \text{ cm}^{-2}$ and a record low X-ray detection limit down to $0.1 \text{ nGyair s}^{-1}$. (Dong et al., Adv. Mater. 2021, 33, e2103078). We can see that the high sensitivity obtained in the relevant literatures because of their high ut product more than $1.2 \times 10^{-2} \text{ cm}^2 \text{ V}^{-1}$, which is much higher than the ut product of $1.8 \times 10^{-4} \text{ cm}^{-2} \text{ V}^{-1}$ in this work. So how can such a low ut product achieve such high detection sensitivity?
2. The sensitivity values might be overestimated in this work. Can the authors do a calculation to estimate the theoretical sensitivity maximum of a semiconductor detector by assuming all X-ray is stopped and all generated charges are extracted. It is possible that a photoconductive gain present, but generally a large density of defects is needed for photoconductive gain. The gain also amplify noise, and thus won't really improve the signal-noise ratio or benefit X-ray detection. In addition, such defects need to elongate the response time, something not desired either.
3. The low detection limit is highly relevant to the spectra of X-ray and the author should provide the spectra of X-ray. It is better for the authors to measure the low detection limit by the standard X-ray spectra such as RQA3 or RQA5.
4. The X-ray imaging of the F-1D TA4Pb3I10 perovskite tablet is performed by the mini-panel linear array scanning system. In this case, the MTF of the detector is overestimated and the characterization method of MTF is very unprofessional. The authors should integrate the perovskite tablet with the CMOS or TFT ROIC and then perform the MTF measurement.
5. The experiment description of process and sample information is barely enough to make evaluation. Sample thickness needs to be described in every figure if they are different.
6. The high sensitivity of F-1D TA4Pb3I10 perovskite tablet is derived from a small range of dose rate less than 35 nGyair s^{-1} as shown in Supplementary Fig. 18. What is the linear dynamic range of the detector? If the linear dynamic range is too small, it would be not very useful for real applications due to different patient body shapes and density of organs.
7. In line 346, "We first polished the F-1D TA4Pb3I10 perovskite tablet with water for further electrode deposition..." It is necessary to clarify the thickness of the TA4Pb3I10 perovskite tablet to get the high sensitivity of $2.5 \times 10^6 \mu\text{C Gyair}^{-1} \text{ cm}^{-2}$. If different thickness of tablets have been used, it needs to be described clearly in the work since the thickness of tablets affects sensitivity values.
8. The reviewer noted that the authors prepared perovskite single crystals first and then made polycrystalline perovskite tablets. As we all know, the performance of semiconductor

single crystals is much higher than that of the polycrystals, why the author does not directly use single crystals to fabricate detectors, but use polycrystal tablets.

Review for the submission: **Water-stable Perovskite X-ray Flat Mini-panel Imager** by Wanting Pan et al.

Wanting Pan et al. shows interesting work on sintered 1D/2D perovskites. On the synthesis I cannot comment, as this part of the paper needs to be reviewed by a specialized chemist.

BUT the X-ray characterization is the weak and questionable part of this paper.

The authors report for the so called 'F-1D' tablets (you mean a pellet right?) on a sensitivity of $2.5E6 \mu\text{C Gy}_{\text{air}}^{-1} \text{cm}^{-2}$?

I'm sorry to tell - but such a sensitivity is *unphysical for an asymmetrically contacted device*. The theoretical sensitivity is given by:

$$\text{X-ray sensitivity} = \frac{e}{\left(\frac{\mu}{\rho}\right)_{\text{air}} W_{\pm}}, \quad (1)$$

where e is the elementary charge, W_{\pm} is electron-hole pair creation energy by ionization, $\left(\frac{\mu}{\rho}\right)_{\text{air}}$ - X-ray mass energy absorption coefficient in the air (X-ray energy dependent). Formula (1) assumes monoenergetic X-ray photons. Here a screenshot of a Matcad session utilizing Eq. 1 for the reported 1D and 2D perovskite with the respective bandgaps.

The screenshot shows the following calculations:

- $\epsilon_{1_bandgap} := 2.2 \cdot e_c \cdot V$
- $\epsilon_{2_bandgap} := 2.8 \cdot e_c \cdot V$
- $\mu_{\rho 1} := 0.024 \frac{\text{cm}^2}{\text{gm}}$
- $\epsilon_{1_pair} := 2.8 \cdot \epsilon_{1_bandgap}$
- $\epsilon_{2_pair} := 2.8 \cdot \epsilon_{2_bandgap}$
- $\text{Sensitivity}_{theory} := \frac{e_c}{\epsilon_{1_pair} \cdot \mu_{\rho 1}} = (6.764 \cdot 10^3) \frac{\mu\text{C}}{\text{cm}^2 \cdot \text{Gy}}$
- $\text{Sensitivity}_{theory} := \frac{e_c}{\epsilon_{2_pair} \cdot \mu_{\rho 1}} = (5.315 \cdot 10^3) \frac{\mu\text{C}}{\text{cm}^2 \cdot \text{Gy}}$

Here we see that theoretical sensitivity (*which is the utmost achievable value*) is between 5.3-6.7E3 $\mu\text{C}/\text{cm}^2/\text{Gy}$. Numbers around 4000 $\mu\text{C}/\text{cm}^2/\text{Gy}$ are typically seen for state-of-the-art commercial CZT detectors! You are reporting on a 1000times higher value?! There must be a serious measurement error! Also Fig. 4(g) and (h) do not have units on the y-axis and are therefore useless. On the positive side I must acknowledge the work in the direction of an imager.

Besides this there is no detailed description of the setup and used X-ray source?

I'm sorry to tell that in the current state, this paper is not suitable for a publication in the nature group. I would look forward for a more specialized journal.

Responses Letter to Reviewers

Reviewer 1

Comment 1: *Wanting Pan et al. shows interesting work on sintered 1D/2D perovskites. On the synthesis I cannot comment, as this part of the paper needs to be reviewed by a specialized chemist.*

BUT the X-ray characterization is the weak and questionable part of this paper.

Response 1: Thanks for the reviewer's comments. We have carefully followed the reviewer's suggestion to revise our manuscript, and more details are discussed in our X-ray characterization part. The point-by-point responses are also attached below.

Comment 2: *The authors report for the so called 'F-1D' tablets (you mean a pellet right?) on a sensitivity of $2.5E6 \mu\text{C Gy}_{\text{air}}^{-1} \text{cm}^{-2}$?*

Response 2: Thanks for your comments. The F-1D tablet could be called a wafer or pellet, as it was made by pressing the perovskite powders (Gebhard J. Matt et al., *Nature Photonics* 2017, 11, 436; Tang et al., *Nature Communications* 2019, 10, 1989).

Regarding the high sensitivity, we calculate the photoconductor gain of our detector in this manuscript. And to make it more clear, we summarized the sensitivity performance of perovskite-based X-ray detectors mentioned in previously reported work with considerable recognition, and the corresponding gain are compared as shown in Figure R1.

Comment 3: *I sorry to tell - but such a sensitivity is unphysical for an asymmetrically contacted device. The theoretical sensitivity is given by:*

$$X\text{-ray sensitivity} = \frac{e}{\left(\frac{\mu}{\rho}\right)_{\text{air}} W_{\pm}}, \quad (1)$$

where e is the elementary charge, W_{\pm} is electron-hole pair creation energy by ionization, $\left(\frac{\mu}{\rho}\right)_{\text{air}}$ – X-ray mass energy absorption coefficient in the air (X-ray energy dependent). Formula (1) assumes monoenergetic X-ray photons. Here a screenshot of a matcad session utilizing Eq. 1 for the reported 1D and 2D perovskite with the respective bandgaps.

$$\begin{aligned} \epsilon_{1_{\text{bandgap}}} &= 2.2 e_c \cdot V & \epsilon_{2_{\text{bandgap}}} &= 2.8 e_c \cdot V \\ \mu_{\rho 1} &= 0.024 \frac{\text{cm}^2}{\text{gm}} & & \\ \epsilon_{1_{\text{pair}}} &= 2.8 \cdot \epsilon_{1_{\text{bandgap}}} & \epsilon_{2_{\text{pair}}} &= 2.8 \cdot \epsilon_{2_{\text{bandgap}}} \\ \text{Sensitivity}_{\text{theory}} &= \frac{e_c}{\epsilon_{1_{\text{pair}}} \cdot \mu_{\rho 1}} = (6.764 \cdot 10^3) \frac{\mu\text{C}}{\text{cm}^2 \cdot \text{Gy}} & \text{Sensitivity}_{\text{theory}} &= \frac{e_c}{\epsilon_{2_{\text{pair}}} \cdot \mu_{\rho 1}} = (5.315 \cdot 10^3) \frac{\mu\text{C}}{\text{cm}^2 \cdot \text{Gy}} \end{aligned}$$

Here we see that theoretical sensitivity (which is the upmost achievable value) is between $5.3\text{-}6.7\text{E}3 \mu\text{C}/\text{cm}^2/\text{Gy}$. Numbers around $4000 \mu\text{C}/\text{cm}^2/\text{Gy}$ are typically seen for state-of-the-art commercial CZT detectors! You are reporting on a 1000 times higher value?! There must be a serious measurement error! Also Fig. 4(g) and (h) do not have units on the y-axis and are therefore useless. On the positive side I must acknowledge the work in the direction of an imager.

Response 3: Thanks for the reviewer's comments, and we appreciate your acknowledgement of the X-ray imaging part. And we have followed the reviewer's suggestion to add the units on the y-axis in Figures 4g and 4h.

Actually, the high sensitivity/photoconductor gain has been widely reported in perovskite X-ray detectors (*Nat. Photonics*, 2017, 11, 315; *Angew. Chem. Int. Ed.*, 2023, 62, e202303445; *Adv. Mater.*, 2023, 2210878; *Nat. Photonics*, 2017, 11, 726). The photoconductor gain stems from the electron injection of the external circuit, and the trapped electron-induced hole injection mechanism allows several orders of signal magnification. Some reports on the high photoconductor gain of perovskite X-ray detectors are listed in Figure R1b.

Figure R1: Theoretical sensitivity (bottom panel in Figure R1a), measured sensitivity (top panel in Figure R1a), and corresponding photoconductor gain factor (Figure R1b). The x-axis of Figure R1a refers to the reference number of the References below.

Table R1: Parameters for theoretical sensitivity calculation.

material	Bandgap (eV)	X-ray energy (keV)	X-ray mass energy absorption coefficient	Reference
----------	--------------	--------------------	--	-----------

			(cm ² /g)	
MAPbI ₃ SC	1.55	22	0.392	Ref. 1
MAPbBr ₃ SC	2.30	8	9.446	Ref. 2
CsFAMA:Sr SC	1.55	60	0.030	Ref. 3
(NH ₄) ₃ Bi ₂ I ₉ SC	2.05	50	0.041	Ref. 4
Cs ₂ AgBiBr ₆ SC	2.00	30	0.154	Ref. 5
Cs ₂ AgBiBr ₆ wafer	2.00	30	0.154	Ref. 6
MA ₃ Bi ₂ I ₉ wafer	2.08	45	0.050	Ref. 7
MAPbI ₃ wafer	1.55	70	0.025	Ref. 8
TA ₄ Pb ₃ I ₁₀ tablet	2.80	120	0.024	this work

The references refer to:

- Ref. 1: Dong et al., *Adv. Materials* 2021, 33, 2103078;
 Ref. 2: Huang et al., *Nature Photonics* 2017, 11, 315-327;
 Ref. 3: Shen et al., *Nature Photonics* 2022, 16, 575;
 Ref. 4: Yang et al., *Nature Photonics* 2019, 13, 602-608;
 Ref. 5: Tang et al., *Nature Photonics* 2017, 11, 726-732
 Ref. 6: Tang et al., *Nature Communications* 2019, 10, 1989;
 Ref. 7: Zhang et al., *Advanced Materials* 2020, 32, 2001981;
 Ref. 8: Gebhars J. Matt et al., *Nature Photonics* 2017, 11, 436-440.

Comment 4: Besides this there is no detailed description of the setup and used X-ray source? I'm sorry to tell that in the current sate, this paper is not suitable for a publication in the nature group. I would look forward for a more specialized journal.

Response 4: We have followed the reviewer's suggestion to add more discussions on this part in the revised manuscript. The X-ray tube with a tungsten target is MGI 320 instrument made by YXLON. The X-ray focal size is controllable from 0.4 to 1 mm, all X-ray experiments in this work were tested under the focal size of 1 mm. The dose rate of the continuous X-ray can be gradually tuned by tube voltage and tube current, and the tube voltage of the X-ray source is fixed at 120 kVp, dose rate of the X-ray is tuned by changing the tube current (from 0.01 mA to 15 mA). A Pb plate with a thickness of ~6 mm is applied to further attenuate the X-ray dose rate to target low-dosage imaging applications. Before we perform the X-ray characterization, we carefully calibrated the dose rate of the X-ray source at every tube current with IBA MagicMaX Multi-Detector (XR 40-150 kV). A Keithley 2400 was used to apply a bias voltage (0~200 V). A DC power KXN-10002D (ZHAOXIN) was used to apply the bias voltage between 200 V~1000 V, and the low-noise current preamplifier (SR570, Stanford Research Systems) and SR830 lock-in amplifier were used to record the signals.

Reviewer 2

Comment 1: The authors show transition between (1D) tryptamine lead iodine ($TA_4Pb_3I_{10}$) and 2D TA_2PbI_4 perovskite tablets. Such transition was made possible because of interplay of water/moisture, cation- π and π - π transitions. The discussed chemistry, and how it influences the electronic band structure is interesting. Subsequently, they show how the water-stable perovskites can be used for X-ray detection.

The work is compelling and can be published in Nature Communication. However, a few clarifications are required before that.

Response 1: We appreciate the reviewer's recognition of our work as well as the suggestion of publication. We have carefully followed the reviewer's suggestion to revise our manuscript and supporting information. Point-by-point responses are listed below.

Comment 2: Clarification on structure.

(a) What exactly is the crystal structure for the 1D case? Does it qualify to be called as perovskite structure?

(b) After water soaking experiments (water-stable perovskites), does the structure contain stoichiometric amount of water in the crystal? How does water is incorporated in the sample? Also refer to very relevant paper: "Releasable Water Charge-Trapping and Water-Resistant Photodetection Using 1d Perovskitoid Hydrate Single Crystal. Adv. Mater. 2022, 34, 2204710."

(c) Authors are suggested to carry out TGA and DSC measurements.

(d) Can author provide the check-cif files in supporting information? What is the status of A-, B-, C-alerts?

Response 2: Thanks for the reviewer's valuable comments and suggestions.

(a) In the unit cell of the 1D perovskite (Figure R2a), three octahedrons are connected in turn by face contact and edge contact, forming the repeat unit of the inorganic part, and between these two columns of the inorganic octahedron, four TA molecules are divided into two parallel groups, with each group of the parallel molecules arranged in reverse order. Perovskite is really a big family that includes continuous/discontinuous $[PbI_6]^{4-}$ octahedrons. Therefore, $TA_4Pb_3I_{10}$ belongs to the low-dimensional 1D perovskites.

Figure R2. Crystal structure of the 1D $TA_4Pb_3I_{10}$ perovskite. (a) exhibits the unit cell

of the crystal. (b, c, d) show the crystal structure from the side of a, b, c, respectively.

(b) Based on our single-crystal data, IR, and TGA experiments, we didn't observe water in the crystal sample. The difference between the mentioned paper (*Adv. Mater.* 2022, 34, 2204710) and this work is discussed in the revised manuscript.

(c) Actually, we already include the TGA measurement (Figure S14) in our first version. The F-1D tablet showed no weight loss before reaching its decomposition temperature of 242°C. Based on these two experiments, we considered that due to the hydrophobic force of the cation- π interactions, the organic molecule TA is driven towards a more closely packed arrangement when soaked into/immersed in the water, and then behaved as the 1D crystal structure. No meaningful information is observed from the DSC tests due to the sensitivity issue of this technique.

(d) Thanks for the reviewer's suggestion, we have added the following check-cif files in supporting information. There is no A- or B- alerts, and the C- alerts include ABSTY02_ALERT_1_C, RADNM01_ALERT_1_C, and PLAT420_ALERT_2_C.

Comment 3: Clarification on charge transport.

(a) Do authors suggest that the hole transport through the organic part and electron transport through the inorganic part? Refer to the very relevant paper "Combining π -Conjugation and Cation- π Interaction for Water Stable and Photoconductive One-Dimensional Hybrid Lead Bromide *J. Phys. Chem. Lett.* 2023, 14, 1870–1876".

(b) Do the water molecules influence charge transport? Refer: "*Adv. Mater.* 2022, 34, 2204710."

(c) I would expect poor carrier mobility from the present hybrid 1D structure, compared to 3D hybrid Pb-I based perovskites, reducing the X-ray detection efficiency of the 1D structure. Any comment?

Response 3: Thanks for your comments.

(a) According to the DFT calculation results, the band edge of the 1D TA₄Pb₃I₁₀ is constructed by the organic TA molecule (VBM) and inorganic Pb-I octahedron (CBM). The organic part should have a major contribution to the hole transport, while the inorganic part has a major contribution to the electron transport.

(b) Water molecules are not included in the crystal structure, and the difference between the mentioned paper (*Adv. Mater.* 2022, 34, 2204710) and this work is discussed in the revised manuscript.

(c) We agree that the 1D TA₄Pb₃I₁₀ perovskite has lower charge carrier mobility compared to the 3D perovskites. While this 1D perovskite has superior stability upon a large applied electric field (E), even at 8000V/cm condition. The charge collection efficiency is actually determined by the $\mu\tau E$ product (or Schubweg distance), which is actually the real Figure-of-Merit to determine the charge collection efficiency/signal/sensitivity. Therefore, we can still achieve a large X-ray sensitivity comparable to the state-of-the-art perovskite X-ray detectors.

Comment 4: To me, the larger part of the manuscript is based on controlling/switching the structure-property relationship, by controlling various intermolecular interactions.

The title of the manuscript does not reflect that part. The title highlights the device part, which is important but probably a smaller contribution in the manuscript.

Response 4: We follow the reviewer's suggestion to revise the title.

Comment 5: *There are other similar approaches to improve the water stability of hybrid perovskites. Refer to: Adv. Energy Mater. 2015, 5, 1501066; ACS Omega 2020, 5, 46, 29631; J. Am. Chem. Soc. 2021, 143, 19901; and so on. Authors might mention those approaches in the revised manuscript, comparing the efficacy of their present approach.*

Response 5: We follow the reviewer's suggestion to highlight the above papers in the role of improving the water stability of hybrid perovskite in the revised manuscript.

Reviewer 3

Comment 1: In this work, the authors reported a water-stable one-dimensional (1D) tryptamine lead iodine ($TA_4Pb_3I_{10}$) perovskite tablet, which can be switched to 2D TA_2PbI_4 perovskite tablet through thermal annealing or water-soaking treatments. This is an interesting discovery, however the understanding of mechanism of this phenomenon is insufficient to understand the device performance. The reported product of the best tablet in this work is $1.8 \times 10^{-4} \text{ cm}^{-2} \text{ V}^{-1}$ is not very good, and not significantly better than other best reported single crystals, such as $MAPbI_3$, $MAPbBr_3$, $FAPbBr_3$, and $FAPbI_3$ etc. It is not clear to this reviewer why these tablets would have much higher sensitivity of $2.5 \times 10^6 \mu\text{C Gy}_{\text{air}}^{-1} \text{ cm}^{-2}$, low detection limit of $5 \text{ nGy}_{\text{air}} \text{ s}^{-1}$ and high spatial resolution of 17.2 lp/mm , or it is just because inappropriate characterizations/claims. Considering the above points and the following concerns, I don't recommend publishing this work in this journal. Below are some detailed comments. The concerns/questions are listed below:

Response 1:

(a) The mechanism of the dimension transition, and corresponding effect on device performance.

In this work, as shown in Figures 1e, 1g, and Figure S7, we discussed that during the tableting process, the external pressure introduced internal stress in the as-prepared perovskite tablet bulk material, and thus causing an enlarged strain in the lattice structure of the as-prepared tablet (1D powder were pressed to be the as prepared tablet (A-1D) tablet process in Figure 1e), leading the A-1D tablet attached with an uncomfortable arrangement. And we schematic the process as depicted in Figure S6. The A-1D tablet with enlarged microstrain will transform to the T-2D tablet with reduced microstrain during thermal annealing (Figures 1d and 1e), since the thermal energy input assists the molecules' micro-vibration to alleviate their uncomfortable arrangement, and then it manifests as transformed 2D tablet (T-2D tablet). The hydrophobic interaction from cation- π interaction drives the tablet to stabilize at the F-1D tablet state with the released microstrain. We schematic this process as exhibited in Figure 1g.

In addition, as shown in Figures 3e and 3g, compared with the A-1D tablet, the F-1D tablet which experienced the whole dimension transition process shows more uniform and pin-hole-free morphologies, giving superior charge collection efficiency/sensitivity. In conclusion, compared with an A-1D tablet, the F-1D tablet showed released microstrain with no pinholes and exhibited better device performance/sensitivity as an X-ray detector.

Dimension transition is a novel strategy to improve the device performance by regulating the strain and pinhole, and related references are listed below:

(1, strain effect on device performance: Sang II Seok et al., *Science* 2020, 370, 108-112; Edward H. Sargent et al., *Nature Communications* 2020, 11, 1514; Bi et al., *Advanced Energy Materials* 2021, 11, 2101018.

2, Pinhole effect on performance: Sang II Seok et al., *Nature* 2023, 616, 724-730; Han et al., *Nature Communications* 2020, 11, 2678; Zhao et al., *Nature Communications*

2016, 7, 12305.)

(b) Explanation of the relationship between $\mu\tau$ and sensitivity.

The reviewer may have a wrong understanding of the $\mu\tau$ product, which mislead his conclusion on the device sensitivity. The physical meaning of this $\mu\tau$ product is the carrier drift length per unit electric field. It is true that a higher $\mu\tau$ product often leads to a larger device sensitivity. However, $\mu\tau E$ product, where E is the external applied electric field, is actually the real Figure-of-the-Merit to determine the charge collection efficiency/sensitivity. It should be noted that we have applied a 1000 V bias on our 1.25 mm tablet device with an electric field of 8000 V/cm. The $\mu\tau E$ product in this work is comparable with the state-of-the-art perovskite X-ray detectors, making it possible to achieve such high sensitivity. More detailed discussions were discussed in the following **Responses**. Besides, the sensitivity in this work was obtained after multiple experiments, and we believe that the sensitivity is measured in a correct manner.

***Comment 2:** The authors reported a high sensitivity of $2.5 \times 10^6 \mu\text{C Gy}_{\text{air}}^{-1} \text{cm}^{-2}$. Similar X-ray sensitivity have already been reported by other methods or compositions. For example, Shen et al. reported a sensitivity of $(2.5 \pm 0.2) \times 10^6 \mu\text{C Gy}_{\text{air}}^{-1} \text{cm}^{-2}$ (Shen et al., Nat. Photonics 2022, 16, 575), Dong et al. reported a higher sensitivity of $5.2 \times 10^6 \mu\text{C Gy}_{\text{air}}^{-1} \text{cm}^{-2}$ and a record low X-ray detection limit down to $0.1 \text{ nGy}_{\text{air}} \text{ s}^{-1}$. (Dong et al., Adv. Mater. 2021, 33, 2103078). We can see that the high sensitivity obtained in the relevant literatures because of their high $\mu\tau$ product more than $1.2 \times 10^{-2} \text{ cm}^2 \text{ V}^{-1}$, which is much higher than the $\mu\tau$ product of $1.8 \times 10^{-4} \text{ cm}^2 \text{ V}^{-1}$ in this work. So how can such a low $\mu\tau$ product achieve such high detection sensitivity?*

Response 2: That is a very interesting question on how a low $\mu\tau$ product achieves a high device sensitivity. The reviewer mentions two papers from my University to compare their $\mu\tau$ product/sensitivity with our device performance. However, the reviewer misunderstood the physical meaning of the $\mu\tau$ product. It is true that a higher $\mu\tau$ product often leads to a large device sensitivity, but a large applied electric field (E) can compensate for the disadvantage, which is very common in the X-ray detection field. $\mu\tau E$ product (or schubweg distance), which indicates the mean drift length of the charge carrier, is the most important Figure-of-the-Merit to determine the charge collection efficiency/signal/sensitivity (Schlesinger, T. E. et al., *Materials Science and Engineering: R: Report* 2001, 32, 103-189; G.F. Knoll & D.S. McGregor, *MRS Online Proceedings Library*, 1993, 302, 3-17).

In the case of 3D perovskite, although the device exhibits a high $\mu\tau$ product of $10^3 \sim 10^2 \text{ cm}^2 \text{ V}^{-1}$, the instability of the large electric field limits their $\mu\tau E$ value and device sensitivity. In contrast, in the case of 1D $\text{TA}_4\text{Pb}_3\text{I}_{10}$ perovskite, although the device attains a relatively low $\mu\tau$ product (10^{-4} order of magnitude), the electric field can be stably applied above 8000 V/cm, thus the applied bias greatly contributes to the $\mu\tau E$.

In the above-mentioned work (Shen et al., *Nat. Photonics* 2022, 16, 575), they achieved a sensitivity of $(2.5 \pm 0.2) \times 10^6 \mu\text{C Gy}_{\text{air}}^{-1} \text{cm}^{-2}$ under the applied electric field of 400 V cm^{-1} . The $\mu\tau$ product of $1.29 \times 10^{-2} \text{ cm}^2 \text{ V}^{-1}$ corresponds to a $\mu\tau E$ of 5.16 cm:

$$\mu\tau E = \mu\tau \times E = 1.29 \times 10^{-2} \text{ cm}^2 \text{ V}^{-1} \times 400 \text{ V cm}^{-1} = 5.16 \text{ cm}$$

In another above-mentioned work (Dong et al., *Adv. Mater.* 2021, 33, 2103078),

they achieved a higher sensitivity of $5.2 \times 10^6 \mu\text{C Gy}_{\text{air}}^{-1} \text{cm}^{-2}$ under the applied electric field of 1000 V cm^{-1} . The $\mu\tau$ product of $1.46 \times 10^{-3} \text{ cm}^2 \text{ V}^{-1}$ corresponds to a $\mu\tau E$ of 1.46 cm:

$$\mu\tau E = \mu\tau \times E = 1.46 \times 10^{-3} \text{ cm}^2 \text{ V}^{-1} \times 1000 \text{ V cm}^{-1} = 1.46 \text{ cm}$$

In our case, the sensitivity we achieved is $2.5 \times 10^6 \mu\text{C Gy}_{\text{air}}^{-1} \text{cm}^{-2}$ under an applied electric field of 8000 V cm^{-1} . The $\mu\tau$ product of $1.8 \times 10^{-4} \text{ cm}^2 \text{ V}^{-1}$ corresponds to a $\mu\tau E$ of 1.44 cm, which is comparable to the above-reported values:

$$\mu\tau E = \mu\tau \times E = 1.8 \times 10^{-4} \text{ cm}^2 \text{ V}^{-1} \times 8000 \text{ V cm}^{-1} = 1.44 \text{ cm}$$

It should be also noted that many papers reported that the device sensitivity can be continuously improved by increasing the applied bias (NOT limited to perovskite materials). However, most perovskite devices will degrade/ions migration due to their instability under a large electric field. That is why most perovskite X-ray detectors don't apply large electric fields as amorphous Se. In our case, our instrument has maximized output of 1000 V, which actually is not a limitation of our X-ray detector. As shown in below Figure R3, amorphous Se even applies a huge electric field of over 100000 V/cm (J. W. Boag, *Xeroradiography* 1973, 18, 3-37), since the $\mu\tau$ product of the amorphous Se is only on the order of $10^{-7} \text{ cm}^2 \text{ V}^{-1}$. However, this doesn't influence the commercialization of amorphous Se X-ray detectors in the past decades, although amorphous Se X-ray detectors don't dominate the flat panel imager anymore due to their weak attenuation of X-rays and cost issues. Based on the above analysis, we believe the high sensitivity of our low-dimensional TA-based X-ray detector is reasonable. Since the external applied electric field is a crucial factor in applications, we cannot evaluate a detector performance solely through $\mu\tau$ product without considering the applied electric field.

To avoid any confusion, we have revised our manuscript to further interpret this question in the revised manuscript.

Figure R3. Collection efficiency of negatively (left)/positively (right) charged α -Se over the electric field.

Comment 3: The sensitivity values might be overestimated in this work. Can the authors do a calculation to estimate the theoretical sensitivity maximum of a semiconductor detector by assuming all X-ray is stopped and all generated charges are extracted. It is possible that a photoconductive gain is present, but generally a large density of defects is needed for photoconductive gain. The gain also amplifies noise, and thus won't really improve the signal-noise ratio or benefit X-ray detection. In addition, such defects need to elongate the response time, something not desired either.

Response 3: According to the reviewer's suggestion, we calculated the theoretical sensitivity of the X-ray detector by equation 1:

$$\text{X-ray sensitivity} = \frac{e}{W_{\pm} \times \left(\frac{\mu_{en}}{\rho}\right)} \quad (1)$$

$$W_{\pm} = 2.8 \times \text{bandgap} + 0.5 \text{ (eV)} \quad (2)$$

Where e is the elementary charge, $\frac{\mu_{en}}{\rho}$ is X-ray mass energy-absorption coefficient in air (Figure R4a), W_{\pm} is the amount of radiation energy consumed per electron-hole pair generated in a semiconductor. Bandgap refers to the bandgap of the response material, in $\text{TA}_4\text{Pb}_3\text{I}_{10}$ case, it is 2.8 eV. Thus, the theoretical sensitivity of the F-1D tablet-based detector is $5 \times 10^3 \mu\text{C Gy}_{\text{air}}^{-1} \text{ cm}^{-2}$.

We also agree with the reviewer that the high sensitivity should result from the photoconductive gain, which has been widely observed in perovskite X-ray detectors. We summarized the sensitivity performance and the corresponding photoconductive gain of perovskite-based X-ray detectors mentioned in published reports with considerable recognition, as shown in Figures R4b and R4c. Figure R4d depicts the gain factor of the X-ray detector in this work under various bias voltages.

It should be also noted that the noise of 1D $\text{TA}_4\text{Pb}_3\text{I}_{10}$ perovskite is quite small, and the signal/noise ratio is still large enough to detect low-dose X-rays as shown in Figure 4h. Shallow defects don't always determine the response time, although sometimes they will. In our device, the response time should be not the major concern, since the large electric field in our device can drive the charges to opposite electrodes within a short transit time.

The references in Figure R4 refer to:

- Ref. 1: Dong et al., *Adv. Materials* 2021, 33, 2103078;
- Ref. 2: Huang et al., *Nature Photonics* 2017, 11, 315-327;
- Ref. 3: Shen et al., *Nature Photonics* 2022, 16, 575;
- Ref. 4: Yang et al., *Nature Photonics* 2019, 13, 602-608;
- Ref. 5: Tang et al., *Nature Photonics* 2017, 11, 726-732
- Ref. 6: Tang et al., *Nature Communications* 2019, 10, 1989;
- Ref. 7: Zhang et al., *Advanced Materials* 2020, 32, 2001981;
- Ref. 8: Gebhars J. Matt et al., *Nature Photonics* 2017, 11, 436-440.

Figure R4. Theoretical sensitivity, measured sensitivity, and corresponding photoconductive gain. (a) shows the X-ray mass energy-absorption coefficient in air, obtained from the NIST database. (b) exhibits the theoretical sensitivity and experimental sensitivity of various perovskite-based X-ray detectors. The x-axis of Figure R3b refers to the number of references. (c) plots the gain factor of various X-ray detectors. (d) depicts the gain factor of the X-ray detector in this work, under various applied voltages.

Comment 4: The low detection limit is highly relevant to the spectra of X-ray and the author should provide the spectra of X-ray. It is better for the authors to measure the low detection limit by the standard X-ray spectra such as RQA3 or RQA5.

Response 4: We follow the reviewer's suggestion to provide the spectra of X-rays, which are emitted from a practically industrial flaw detection machine (The X-ray tube with a tungsten target is MGI 320 instrument made by YXLON, 20~160 kV), which is more meaningful than the standard X-ray spectra such as RQA3 or RQA5. The low detection limit in this work can reflect the practical condition in the industry.

Figure R4: The X-ray spectra.

Comment 5: The X-ray imaging of the F-1D TA₄Pb₃I₁₀ perovskite tablet is performed by the mini-panel linear array scanning system. In this case, the MTF of the detector is overestimated and the characterization method of MTF is very unprofessional. The authors should integrate the perovskite tablet with the CMOS or TFT ROIC and then perform the MTF measurement.

Response 5: The X-ray images of the standard X-ray lead bar phantoms are performed by a single pixel scanning, as we mentioned in lines 377-379, to measure the performance limit of the pixel made by the TA-based perovskite material, we used a single pixel to image the standard X-ray lead bar phantoms by 2D displacement platform. And in this case, the characterization method of MTF has been widely used (Nam-Gyu Park et al., *Nature*, 2017, 550, 87-91; Tang et al., *Nature Communications* 2019, 10, 1989; Shen et al., *Nat. Photonics* 2022, 16, 575).

Comment 6: The experiment description of process and sample information is barely enough to make evaluation. Sample thickness needs to be described in every figure if they are different.

Response 6: Thanks for the reviewer's comments. We follow the reviewer's suggestion to add the thickness of the sample in every characterization clearly in the revised manuscript.

Comment 7: The high sensitivity of F-1D TA₄Pb₃I₁₀ perovskite tablet is derived from a small range of dose rate less than 35 nGy_{air} s⁻¹ as shown in Supplementary Fig.18. What is the linear dynamic range of the detector? If the linear dynamic range is too small, it would be not very useful for real applications due to different patient body shapes and density of organs.

Response 7: The device shows an intense X-ray response under tens μGy_{air} s⁻¹ condition for practical medical exams, but perovskite X-ray detectors basically aim to low-dose

detection/imaging to outperform the traditional X-ray detectors due to their large device sensitivity. Therefore, it is actually more meaningful to focus on the low-dose X-ray range.

Comment 8: *In line 346, “We first polished the F-1D $\text{TA}_4\text{Pb}_3\text{I}_{10}$ perovskite tablet with water for further electrode deposition...” It is necessary to clarify the thickness of the $\text{TA}_4\text{Pb}_3\text{I}_{10}$ perovskite tablet to get the high sensitivity of $2.5 \times 10^6 \mu\text{C Gy}_{\text{air}}^{-1} \text{cm}^{-2}$. If different thickness of tablets has been used, it needs to be described clearly in the work since the thickness of tablets affects sensitivity values.*

Response 8: We follow the reviewer’s suggestion to revise the manuscript

Comment 9: *The reviewer noted that the authors prepared perovskite single crystals first and then made polycrystalline perovskite tablets. As we all know, the performance of semiconductor single crystals is much higher than that of the polycrystals, why the author does not directly use single crystals to fabricate detectors, but use polycrystal tablets.*

Response 9: Growing single crystals is often time-consuming with limited dimensional sizes, although the performance is superior to polycrystals counterparts. Polycrystal tablets are more compatible with the industry requirements, and are easily processed in a large area with low cost and good reproducibility.

Reviewers' comments:

Reviewer #3 (Remarks to the Author):

The revised manuscript is better. However, there are still some concerns needed to be addressed before it publishes.

1. It seems that the 1D to 2D transition process is designed to prepare a 1D tablet with less strain stress and high stability. It is a very important procedure for the highly sensitive tablet X-ray detector. However, during the transition, the chemical formulas changed a lot, which means a part of chemical did not exist in the final product. Which kind of state are these extra chemicals in the T-2D perovskite. (TA:Pb:I ratios are 4:3:10 for 1D and 2:1:4 for 2D perovskites.)
2. It is easier to understand that heat will drive the phase/chemical transitions as heat can be transported through the objects. However, it is hard to understand the water induced phase/chemical transition in this research. As the water/sample contact just exists at the tablet surface, the driving force accounts for the tablet phase/chemical transition should be well discussed in this manuscript.
3. There should be anisotropic carrier transport in the low dimensional perovskite, which will affect the charge collection or transport in the final tablet. It is suggested that the authors add the data related to the anisotropic carrier transport properties of single crystal sample as well as tablet sample.
4. As p-i-n structured devices has been build in the manuscript, it is suggested to provide ut products for both electron and hole. At the same time, the authors attribute the gain to the photoconductive gain in the manuscript and response letter. The trapped carrier type (e or h) and transported carrier (e or h) are suggest to be determined in the revised manuscript.
5. As listed in the experimental section, the SR830 locking amplifier has been used to record the signal. Is pulsed X-ray source has been used in the X-ray response test? At the same time, the device area is not provided in the manuscript.

Reviewer #4 (Remarks to the Author):

In the manuscript entitled "Cations- π Interactions Enabled Water-stable Perovskite X-ray Flat Mini-panel Imager" Pan W. et al. report polycrystalline lead halide X-ray detector based on 1D TA4Pb3I10 perovskite, which was not reported for X-ray detection so far. The reported material stability against water and high electrical field is rather appealing, however, the X-ray detection characterization exhibits numerous drawbacks, which questions the author's main claims, particularly regarding X-ray sensitivity and spatial resolution. The paper might be considered for publication in Nature Communication after a major revision of the points below is conducted. Please find my comments in details below:

As mentioned by Reviewers 1 and 3, the X-ray sensitivity seems to be overestimated. The possible reason for overestimation could be found in the description of the utilized X-ray tube provided by the authors. It is mentioned that the X-ray tube has a focal spot of 1 mm (I assume it is a diameter). With such a tiny beam size it is effortless to drastically underestimate the radiation dose due to a mismatch of the beam size to the used dosimeter input window size (which is roughly 87 mm², as can be found here for MagicMaX Multi-Detector (XR 40-150 kV) https://med-renovate.com/wp-content/uploads/2018/10/br-ib-medical-imaging-product-overview_en_rev7_opt.pdf). A typical dosimeter is calibrated for homogeneous over dosimeter area X-ray irradiation. When the smaller area is irradiated, the

output signal is normalized on the whole sensitive dosimeter area, thus showing a much lower value than the actual input dose. I would fear that the authors underestimated the input dose rate on 1 or 2 orders of magnitude, leading to an overestimation of the reported X-ray sensitivity 10 or 100 times. To avoid this issue, I suggest that the authors characterize their device at an out-of-focus X-ray tube distance, which would increase the beam size and allow accurate dose measurements.

As the authors mentioned in the X-ray detection characterization part, the best spatial resolution data (Fig. 5h-k) was obtained with a single-pixel device with a size of 20 μm by scanning. I would strongly recommend not reporting such data for several reasons: 1) single-pixel scanning doesn't have much practical sense for real medical imaging applications (as with scanning time the accumulated dose increases); 2) with small enough pixel sizes any high number of spatial resolution can be obtained with any kind of X-ray sensitive material. A practically relevant and material-characterizing measurement would be a simultaneous readout of several pixels to see the object edge and severity of the current cross-talk between pixels. Such measurements might be performed with the device shown on Fig5. c-e.

For the noise performance assessment, I suggest for the authors to avoid the lowest detectable dose rate metrics, but use the formalism of noise equivalent dose. The noise equivalent dose metrics is more relevant for a medical application since the most important for harm minimization to the medical imaging patient is to minimize integrated dose, but NOT dose rate.

The detector speed isn't reported in the manuscript, while it is important characteristics of an X-ray detector. I suggest for the authors to show the rise and the fall times of a square X-ray shape pulse with their detectors.

Line 32-34: "However, their sensitivities are still not satisfied by considering the applied high dose in X-ray generators and the subsequent damage to the human body".

High sensitivity solely doesn't satisfy requirement for low-dose imaging. The real figure of merit is signal-to-noise ratio normalized on number of X-ray photons quanta (i.e. dose), which is described by Detective Quantum Efficiency metrics. X-ray sensitivity only characterize the signal amplitude, but tells nothing on the detector noise (which could be rather high for material with high sensitivity, caused by photoconductive gain). I suggest for the authors to be comprehensive, when describing requirements of the novel detector development.

Line 82 "the crystal growth at a low crystalline rate of $<50 \text{ mm}^3/\text{min}$ "

I would remove word "crystalline", rewriting like this "the crystal growth at a low crystalline rate $<50 \text{ mm}^3/\text{min}$ ".

Line 328-332.

For determination of W_{\pm} , please cite original C.A. Klein paper
<https://doi.org/10.1063/1.1656484>

Line 328-332.

Please specify which energy of X-ray was assumed for determination of μ_{en}/ρ .

Fig. 5e is confusing, hardly explain readout system and isn't commented in the main text. I suggest for the authors to add more comprehensive description of readout system for imaging.

Reviewer #5 (Remarks to the Author):

Pan et al reported using polycrystalline tablet 1D TA4Pb3I10 for X-ray detection imager application. It's interesting about interchangeable 1D and 2D crystal structures with the presence of water and heat. However, no solid understanding of how material process

results can change the crystal phase and correlate with device performance. Meanwhile, I agreed with Reviewer 1 and Reviewer 3 that many concerns and questions still remained even after the first round of revision. Based on the response and reviewers' comments, some additional comments are listed below.

For the F-1D device after water treatment and then drying the tablet, why do the authors use Cr/BCP/C60/perovskites/Au device structure? It's known that the perovskite single crystals have charge mobility imbalance issues. However, I did not find anything about understanding the F-1D material or devices as Reviewer 1 also has the same concern. I would suggest the author to do some detailed explanation and this will allow readers to understand the device structure design principle.

Regarding the question about single crystal and polycrystalline materials for the device testing, although the authors' response the polycrystalline tablet preparation is easier for industrial adaption, however, I agreed with reviewer 3 that the performance for single crystal is much higher than polycrystalline materials, it should be also tested the single crystal device to understand the best performance of this material.

In Figure 4b, why did the X-ray sensitivity increase in the first 48 hours? As the SNR data in Fig 4c is quite stable, this suggested the response was getting increased. However, I did not find any detailed understanding about the material and device performance correlation in the MS.

For the device stability test in Figure 4g, the rise and decay time of signals seems different over time, this information is important because the device response time is essential for understanding the charge transport and collection in device. However, I did not see the detailed comparison in the MS for the rise and decay time which will be useful information for readers.

Some minor comments

(1) Many figure labelings are missing in SI, which is hard to understand and interpret the data.

(2) Some of the SI figures cited from MS are wrong, such as Figure S10 depicts IR spectra, however, Figure S10 is ^{13}C ss-NMR. The authors should go over MS and SI with correct citations.

Responses Letter to Reviewers

Reviewer 3

Comment 1: The revised manuscript is better. However, there are still some concerns needed to be addressed before it publishes.

It seems that the 1D to 2D transition process is designed to prepare a 1D tablet with less strain stress and high stability. It is a very important procedure for the highly sensitive tablet X-ray detector. However, during the transition, the chemical formulas changed a lot, which means a part of chemical did not exist in the final product. Which kind of state are these extra chemicals in the T-2D perovskite. (TA:Pb:I ratios are 4:3:10 for 1D and 2:1:4 for 2D perovskites.)

Response 1: Thanks for the reviewer's comments, and we appreciate your acknowledgement of the revised manuscript. As claimed in lines 150-152 in the previous version of the revised manuscript, we explained the transition from 1D to 2D tablets with the driving of the relaxation of strained TA^+ . During the thermal energy input process, TA^+ reassembled into a relatively loose packing mode, which will cause the inorganic octahedra part to fit in this new packing mode and finally self-adaptively stabilize at 2D arrangement mode, thus representing the XRD characterization of the 2D structure. However, based on the TGA characterization (Figure R1a), we think that there are no changes in chemical composition during the whole transition process, as the weight turning points of the A-1D, T-2D, and F-1D show no changes. Besides, compared with A-2D (as-prepared 2D perovskite), the XRD peaks of the T-2D show obvious small-angle shift (Figure R1b), representing a much looser packing mode exists in the T-2D tablet, and signifying much weaker intermolecular interactions between molecules. In this case, we speculate that this T-2D structure is inconsistent with A-2D, and we could get the conclusion that the dimension transition process manifested in XRD is dominated by the growth dimension of the inorganic octahedra, and the growth dimension of the inorganic octahedra is significantly affected by the assembly modes of the TA^+ molecules, while the assembly modes could be driven to change through the thermal energy or water treatment, which can drive the change of the molecular conformation and the intermolecular interactions. We have added more discussions in the revised manuscript in blue.

Figure R1: Characterization of TGA (a) and XRD (b) measurements.

Comment 2: It is easier to understand that heat will drive the phase/chemical transitions as heat can be transported through the objects. However, it is hard to understand the water induced phase/chemical transition in this research. As the water/sample contact just exists at the tablet surface, the driving force accounts for the tablet phase/chemical transition should been well discussed in this manuscript.

Response 2: Thanks for your comments. We have added a more detailed discussion of the transition process in the revised manuscript in *blue*. The reason why water can drive the 2D packing mode to 1D mode is because of the available hydrophobic interaction of the TA-based perovskite. The invasion of the water molecules will drive the TA⁺ molecules arranged in a closely packed mode, this packing mode once induced, the molecules reassembly follows, and this reassembly process will not just occur on the surface for the closely packed mode could also affect the neighboring loosely packed part to achieve a comfortable state. That is to say, the driving force will be transmitted after it is induced, thus the water can induce phase/chemical transition.

Comment 3: There should be anisotropic carrier transport in the low dimensional perovskite, which will affect the charge collection or transport in the final tablet. It is suggested that the authors add the data related to the anisotropic carrier transport properties of single crystal sample as well as tablet sample.

Response 3: Thanks for the reviewer's comments. We checked the anisotropic carrier transport properties of the single crystal and the final tablet (F-tablet). The crystal planes of the 1D TA₄Pb₃I₁₀ perovskite are shown in Figure **R2a**. The anisotropic property is performed as exhibited in Figure **R2b**, a best carrier transport property is performed along the extension direction of the octahedra (blank line in Figure **R2b**, expressed as \perp plane (0k0)), the carrier transport along the direction of \perp plane (k00) (red line in Figure **R2b**) comes second, and a relatively poor carrier transport is performed along the direction of \perp plane (00k) (blue line in Figure **R2b**). In addition, the anisotropic property of the F-1D tablet is also confirmed, as shown in Figure **R2c**, we express the loading surface during tableting as in-plane direction, and express the normal direction of the in-plane as out-of-plane. As depicted in Figure **R2d**, a larger carrier transport property can be confirmed along in-plane direction. We have added these anisotropic data in the SI file.

Figure R2: Characterization of the anisotropic carrier transport of 1D single crystal samples (a, b) as well as F-1D tablet samples (c, d). The tube voltage of the X-ray source is 120 kV_p, and the dose rate is 1.01 $\mu\text{Gy s}^{-1}$. The applied bias voltage on devices is 200 V, all the device area is 1 mm², the device thickness of single crystals is 1.2 mm, and the thickness of the tablet device is 0.4 mm.

Comment 4: As *p-i-n* structured devices has been built in the manuscript, it is suggested to provide ut products for both electron and hole. At the same time, the authors attribute the gain to the photoconductive gain in the manuscript and response letter. The trapped carrier type (*e* or *h*) and transported carrier (*e* or *h*) are suggest to be determined in the revised manuscript.

Response 4: Thanks for the reviewers' comments. The $\mu\tau$ products for both electron and hole are supplied in Figure R3a. The $\mu\tau_h$ product of $2.8 \times 10^{-4} \text{ cm}^2 \text{ V}^{-1}$ is slightly higher than the $\mu\tau_e$ product of $2.1 \times 10^{-4} \text{ cm}^2 \text{ V}^{-1}$. Besides, based on the SCLC experiments of the electron-only/hole-only devices (Figure R3b), the trapped carrier type is electron, and the transport carriers are holes. We have followed the reviewer's suggestion to add these discussions in the revised manuscript.

Figure R3: a. Characterization of $\mu\tau_e$ and $\mu\tau_h$. b. SCLC experiments of electron-only and hole-only tablet devices. The device structure of the electron-only device is Cr/C₆₀/perovskite/C₆₀/Cr, and the hole-only device's device structure is Au/MoO₃/perovskite/MoO₃/Au. The thickness of the C₆₀ and MoO₃ is 25 nm, the thickness of electrodes Au and Cr is 30 nm. The tube voltage of the X-ray source is 120 kV_p, the dose rate is 1.01 $\mu\text{Gy s}^{-1}$. the device area is 1 mm² and the device thickness is 0.1 mm.

Comment 5: As listed in the experimental section, the SR830 locking amplifier has been used to record the signal. Is pulsed X-ray source has been used in the X-ray response test? At the same time, the device area is not provided in the manuscript.

Response 5: A lead chopper wheel was used to modulate the X-ray irradiation when we used the SR830 locking amplifier to record the signal. The device area in every experiment is provided in the revised manuscript.

Reviewer 4

Comment 1: *In the manuscript entitled “Cations- π Interactions Enabled Water-stable Perovskite X-ray Flat Mini-panel Imager” Pan W. et al. report polycrystalline lead halide X-ray detector based on 1D $\text{TA}_4\text{Pb}_3\text{I}_{10}$ perovskite, which was not reported for X-ray detection so far. The reported material stability against water and high electrical field is rather appealing, however, the X-ray detection characterization exhibits numerous drawbacks, which questions the author's main claims, particularly regarding X-ray sensitivity and spatial resolution. The paper might be considered for publication in Nature Communication after a major revision of the points below is conducted. Please find my comments in details below:*

Response 1: We appreciate the reviewer's recognition of our work in material stability. We have carefully followed the reviewer's suggestion to revise our manuscript and supporting information. Point-by-point responses are listed below.

Comment 2: *As mentioned by Reviewers 1 and 3, the X-ray sensitivity seems to be overestimated. The possible reason for overestimation could be found in the description of the utilized X-ray tube provided by the authors. It is mentioned that the X-ray tube has a focal spot of 1 mm (I assume it is a diameter). With such a tiny beam size it is effortless to drastically underestimate the radiation dose due to a mismatch of the beam size to the used dosimeter input window size (which is roughly 87 mm², as can be found here for MagicMaX Multi-Detector (XR 40-150 kV) https://med-renovate.com/wp-content/uploads/2018/10/br-ib-medical-imaging-product-overview_en_rev7_opt.pdf). A typical dosimeter is calibrated for homogeneous over dosimeter area X-ray irradiation. When the smaller area is irradiated, the output signal is normalized on the whole sensitive dosimeter area, thus showing a much lower value than the actual input dose. I would fear that the authors underestimated the input dose rate on 1 or 2 orders of magnitude, leading to an overestimation of the reported X-ray sensitivity 10 or 100 times. To avoid this issue, I suggest that the authors characterize their device at an out-of-focus X-ray tube distance, which would increase the beam size and allow accurate dose measurements.*

Response 2: Thanks for the reviewer's comments and suggestions. Actually, when we performed the X-ray experiments, we put the dosimeter/device at an out-of-focus X-ray tube distance. As shown in Figure R4, the focal size is actually the area of the metal target being bombarded by the accelerated electrons. This X-ray equipment is an advanced MGI 320 instrument made by YXLON (around \$100,000), which can tune the focal size to adopt different applications. Since most X-ray tube has no options on this parameter, only a few people are familiar with this. Indeed, we tried to keep a large distance between the X-ray tube and our devices to obtain a uniform X-ray irradiated zone for accurate evaluations, and this irradiated circle area has a diameter of 180 mm, 10~60 times larger than our devices. We have improved our descriptions in the X-ray characterization part in blue.

Figure R4. Diagram of the X-ray radiation source.

Comment 3: As the authors mentioned in the X-ray detection characterization part, the best spatial resolution data (Fig. 5h-k) was obtained with a single-pixel device with a size of 20 μm by scanning. I would strongly recommend not reporting such data for several reasons: 1) single-pixel scanning doesn't have much practical sense for real medical imaging applications (as with scanning time the accumulated dose increases); 2) with small enough pixel sizes any high number of spatial resolution can be obtained with any kind of X-ray sensitive material. A practically relevant and material-characterizing measurement would be a simultaneous readout of several pixels to see the object edge and severity of the current cross-talk between pixels. Such measurements might be performed with the device shown on Fig5. c-e.

Response 3: Thanks a lot for your comments and suggestions. Actually, we couldn't agree with the viewpoint that any kind of X-ray-sensitive material can perform any high number of spatial resolutions. To achieve an X-ray image with high spatial resolution, enough signal current and high sensitivity must be required to meet the minimum signal value that the imaging system can stably capture and to distinguish the weak dose rate changes of the X-ray obscured by the image object. In our case, an X-ray image with high spatial resolution was achieved by the X-ray detector with a small pixel size of 20 μm , this should be not easy to achieve for low-sensitivity materials. Besides, although ignoring the current cross-talk between pixels, single-pixel scanning represents the comprehensive performance of a detector, so we think the X-ray image scanned by the single-pixel should be reported in this work. Actually, high resolution can only be achieved with a tiny device pixel size, but a small device size/area definitely reduces the signal amplitude. So high device sensitivity is a requisite for high-resolution imaging, and our high-resolution result is the best evidence for the device sensitivity.

Comment 4: For the noise performance assessment, I suggest for the authors to avoid the lowest detectable dose rate metrics, but use the formalism of noise equivalent dose. The noise equivalent dose metrics is more relevant for a medical application since the most important for harm minimization to the medical imaging patient is to minimize integrated dose, but NOT dose rate.

Response 4: Thanks for your comments and suggestions. We have carefully read and studied the previously reported works (Sergii Yakunin, Osman M. Bakr, Maksym V. Kovalenko et al., *Nature Photonics*, 2023, 17, 510-517), and tried to characterize the NED of our X-ray detector based on the noise experiments, as shown in Figure R5a. The NED of the F-1D detector can reach 118 nGy, comparable with the reported scintillator detectors 4336WXv2, 4336Rv3, 4336Wv4 of 459 nGy, 666 nGy, and 566 nGy (Isaias D. Job et al., *Conference of SPIE Medical Imaging*, 2016, 9783, 97833T).

Figure R5: Characterization of NED of the F-1D detector. The integration time is 300 ms.

Comment 5: The detector speed isn't reported in the manuscript, while it is important characteristics of an X-ray detector. I suggest for the authors to show the rise and the fall times of a square X-ray shape pulse with their detectors.

Response 5: Thanks for the reviewer's suggestion. We have followed the suggestion to supply the detector speed in Figure R6. Related discussion is also added in the revised manuscript in blue.

Figure R6: Characterization of the response speed of the detector.

Comment 6: Line 32-34: “However, their sensitivities are still not satisfied by considering the applied high dose in X-ray generators and the subsequent damage to the human body”.

High sensitivity solely doesn't satisfy requirement for low-dose imaging. The real figure of merit is signal-to-noise ratio normalized on number of X-ray photons quanta (i.e. dose), which is described by Detective Quantum Efficiency metrics. X-ray sensitivity only characterize the signal amplitude, but tells nothing on the detector noise (which could be rather high for material with high sensitivity, caused by photoconductive gain). I suggest for the authors to be comprehensive, when describing requirements of the novel detector development.

Response 6: Thanks for the reviewer's suggestions. We have followed the suggestion to improve the description of “However, their sensitivities are still not satisfied by considering the applied high dose in X-ray generators and the subsequent damage to the human body” in blue in the revised manuscript.

About the figure of merit of DQE, due to the limitation of our system, we can't obtain this accurate data now. In our subsequent works, we will take this figure of merit into account.

Comment 7: Line 82 “the crystal growth at a low crystalline rate of <math><50 \text{ mm}^3/\text{min}</math>” I would remove word “crystalline”, rewriting like this “the crystal growth at a low crystalline rate <math><50 \text{ mm}^3/\text{min}</math>”.

Line 328-332.

Response 7: Thanks for the reviewer's suggestions. We have followed the reviewer's suggestion and rewritten the sentence as “the crystal growth at a low crystalline rate <math><50 \text{ mm}^3/\text{min}</math>”.

Comment 8: For determination of W_{\pm} , please cite original C.A. Klein paper <https://doi.org/10.1063/1.1656484>

Line 328-332.

Response 8: Thanks for the reviewer's suggestion. We have added the citation of the original C.A. Klein paper in blue.

Comment 9: Please specify which energy of X-ray was assumed for determination of μ_{en}/ρ .

Response 9: Thanks for the reviewer's suggestion. X-ray with the energy of 120 keV was assumed for the determination of μ_{en}/ρ , and we have added related descriptions in blue in the revised manuscript.

Comment 10: Fig. 5e is confusing, hardly explain readout system and isn't commented in the main text. I suggest for the authors to add more comprehensive description of readout system for imaging.

Response 10: Thanks for the reviewer's suggestion. We have added more descriptions of the readout system in blue in the revised manuscript.

Reviewer 5

Comment 1: Pan et al reported using polycrystalline tablet 1D $\text{TA}_4\text{Pb}_3\text{I}_{10}$ for X-ray detection imager application. It's interesting about interchangeable 1D and 2D crystal structures with the presence of water and heat. However, no solid understanding of how material process results can change the crystal phase and correlate with device performance. Meanwhile, I agreed with Reviewer 1 and Reviewer 3 that many concerns and questions still remained even after the first round of revision. Based on the response and reviewers' comments, some additional comments are listed below.

Response 1: Thanks for the reviewers' comments. We have carefully followed the reviewer's suggestion to revise our manuscript and supporting information. Point-by-point responses are listed below.

Comment 2: For the F-1D device after water treatment and then drying the tablet, why do the authors use Cr/BCP/C60/perovskites/Au device structure? It's known that the perovskite single crystals have charge mobility imbalance issues. However, I did not find anything about understanding the F-1D material or devices as Reviewer 1 also has the same concern. I would suggest the author to do some detailed explanation and this will allow readers to understand the device structure design principle.

Response 2: Thanks for the reviewer's comment.

- 1) We use the device structure of Cr/BCP/C₆₀/perovskites/Au to suppress the dark leakage current and extract the photogenerated carriers, as shown in Figures **R7a** and **R7b**. The photoconductive device with gain is often accompanied by a large dark current. To suppress the dark current, we designed the C₆₀/BCP thin layer between perovskite and electrode Cr as the blocking layer of the injection of external electrons under dark conditions. The perovskite layer will generate free charges upon X-ray irradiation, and part of the generated electrons will be trapped by shallow defects to induce a photoconductive gain, which results in a high device sensitivity. We have added this explanation in the method part in *blue*. The mechanism of these photoconductive gain devices has been repeatedly reported in previous works (Huang et al., *Advanced Optical Materials*, 2014, 2, 549-554; Jonas Kublitski, Karl Leo et al., *Nature Communications*, 2021, 12, 4259).
- 2) Based on the characterization of $\mu\tau_e$ and $\mu\tau_h$ of our device, no obvious charge imbalance issues exist in our device. As shown in Figure **R7c**, the $\mu\tau_e$ of $2.1 \times 10^{-4} \text{ cm}^2 \text{ V}^{-1}$ is just slightly lower than $\mu\tau_h$ of $2.8 \times 10^{-4} \text{ cm}^2 \text{ V}^{-1}$, while in the CdZnTe detector case, $\mu\tau_e$ of $6.1 \times 10^{-3} \text{ cm}^2 \text{ V}^{-1}$ is over one order of magnitude larger than the $\mu\tau_h$ of $3.1 \times 10^{-4} \text{ cm}^2 \text{ V}^{-1}$ (Figure **R7d**) (J. E. Toney, et. al., *IEEE Transactions on Nuclear Science* 1998, 45, 105-113; Schlesinger, T. E. et al., *Materials Science and Engineering: R: Report* 2001, 32, 103-189).

Figure R7: Diagram of the device structure (a, b) and characterization of $\mu\tau_e$ and $\mu\tau_h$ (c). $\mu\tau_e$ and $\mu\tau_h$ of the reported CdZnTe detector (d).

Comment 3: Regarding the question about single crystal and polycrystalline materials for the device testing, although the authors' response the polycrystalline tablet preparation is easier for industrial adaption, however, I agreed with reviewer 3 that the performance for single crystal is much higher than polycrystalline materials, it should be also tested the single crystal device to understand the best performance of this material.

Response 3: Thanks for your comments. We characterized the sensitivity of the single crystal in three planes under various electric field intensities, as exhibited in Figure R8. The sensitivity of the single crystal is higher than the tablet until the bias voltage is large enough to exceed 660 V/mm. We have added this result in the supporting information.

Figure R8: X-ray sensitivity characterization of the $\text{TA}_4\text{Pb}_3\text{I}_{10}$ single crystal. The tube voltage of the X-ray source is 120 kV_p, the device area is 1 mm², the device thickness of the single crystal is 1.2 mm, and the tablet data is previously reported in Figure 4f in

our MS.

Comment 4: In Figure 4b, why did the X-ray sensitivity increase in the first 48 hours? As the SNR data in Fig 4c is quite stable, this suggested the response was getting increased. However, I did not find any detailed understanding about the material and device performance correlation in the MS.

Response 4: Thanks for the reviewer's comments.

1) SNR is calculated by the formula of $SNR = \frac{I_{signal}}{I_{noise}} = \frac{I_{photo} - \overline{I_{photo}}}{\sqrt{\frac{1}{N} \sum_i^N (I_i - \overline{I_{photo}})^2}}$, where

includes the variance of this set of signal data, resulting that the SNR result is seriously affected by the dispersion degree of the calculated data. Thus, SNR is often used to indicate the detection limit as IUPAC defined rather than detection capability.

2) The correlation between the material and device performance can be referred to the Figures 1e and 1g. As claimed in the revised MS in lines 132-134, several cycles of dimension transitions will completely relax the microstrain of the tablet material, serving as ideal large-area X-ray detectors.

Comment 5: For the device stability test in Figure 4g, the rise and decay time of signals seems different over time, this information is important because the device response time is essential for understanding the charge transport and collection in device. However, I did not see the detailed comparison in the MS for the rise and decay time which will be useful information for readers.

Response 5: Thanks for the reviewer's comments. For the device stability test in Figure 4g, the rise and decay time of the signal were actually limited by the shutter of the X-ray tube, which was manually controlled. We have supplied the response time by employing a fast chopper in Figure R9, and we added this data in the revised SI file.

Figure R9: Characterization of the response time of the device.

Comment 6: *Many figure labelings are missing in SI, which is hard to understand and interpret the data.*

Response 6: Thanks for the reviewer's suggestion. We have followed the suggestion to seriously check these kinds of problems in the revised manuscript.

Comment 7: *Some of the SI figures cited from MS are wrong, such as Figure S10 depicts IR spectra, however, Figure S10 is ^{13}C ss-NMR. The authors should go over MS and SI with correct citations.*

Response 7: Thanks for the reviewer's suggestion. We have followed the suggestion to seriously correct them in the revised manuscript.

REVIEWER COMMENTS

Reviewer #3 (Remarks to the Author):

The revised manuscript is better now. However, as mentioned by reviewers 3, 4 and 5, there are still some important concerns and questions remained in this revision, which are needed to be addressed before it can be considered for publication. The detail comments are below:

1. As mentioned by three reviewers, the X-ray sensitivity seems to be overestimated. Even though the authors explained the structure of the X-ray source used and the manufacturing company or the price, it does not indicate whether the dose rate was underestimated during the measurements. In addition to the underestimation of the dose rate, inaccurate measurement of the detector area can also lead to overestimation of sensitivity.
2. The authors still have not explained exactly why this 1D perovskite can achieve such a surprisingly high sensitivity, which is the main reason that many reviewers have been skeptical about the sensitivity so far.
3. The reversible transition between 1D and 2D perovskites is an interesting phenomenon, but the authors do not explain exactly why this transition can occur reversible many times. I agree with reviewer 3, during the transition, the chemical formulas changed a lot, which means a part of chemical did not exist in the final product. Although the author tries to explain this reason through thermal analysis, it is not convincing. Because part of the chemical component is lost after the sample is heated, the sample does not have the conditions for reversible phase transition.
4. In the revision, the authors measured the response time. However, limited information is given on the time response speed, as it is suggested that photoconductive gain and utE plays a significant role, it is necessary to comment its effect on the rise/fall time.
5. In Figure S20, the authors obtained sensitivity by fitting the dose rate-dependent current-density of the detector. However, it seems that when the dose rate is 0, the detector still shows current density, which is very strange.
6. I agree with the reviewer 4, I would strongly recommend not reporting such spatial resolution from the single-pixel device, because it doesn't make sense in real X-ray imaging applications.

Reviewer #4 (Remarks to the Author):

I appreciate the changes that the authors made to improve the manuscript. In my opinion, it might be published in the Nature Communication journal

Reviewer #5 (Remarks to the Author):

The authors have addressed most of the comments from the previous review, however, I found some questions for ss-NMR data as below.

In the ss-NMR data in Fig 3b, the author used the 2D $^{13}\text{C}\{^1\text{H}\}$ HETCOR to characterize the cross peaks and correlation between C and H to further confirm the structure, however, the author states that the solid-state ^1H spectra of both structures do not distinguish peaks of hydrogen atoms in different chemical environments. Thus, the HETCOR characterization for understanding which proton is attached to which C in a given molecule. However, it is not clear and can not provide much useful information.

The author further used NOESY to understand the intermolecular interaction with water. It's interesting to see the chemical shift after titration with D₂O in ACN. However, why the authors change the D-solvent system from DMSO to ACN? From FigS12, the peaks in DMSO as solvent is much sharper than in ACN system. I suggest the authors also present the TAI-Pbl₂ in DMOS with D₂O titration experiments to double-confirm the results. I think this part is much useful than 2D HETCOR data in Fig3b, I would suggest removing HETCOR data to SI and enlarging the NOESY data with details characterizations.

In addition, the labeling in Figure 3b, 3c and figS12 are different but the discussion is on the TAI molecules, which is confusing when reading. I would suggest the author to keep them consistent in the discussion.

Responses Letter to Reviewers

Reviewer 3

***Comment 1:** The revised manuscript is better now. However, as mentioned by reviewers 3, 4 and 5, there are still some important concerns and questions remained in this revision, which are needed to be addressed before it can be considered for publication. The detailed comments are below:*

As mentioned by three reviewers, the X-ray sensitivity seems to be overestimated. Even though the authors explained the structure of the X-ray source used and the manufacturing company or the price, it does not indicate whether the dose rate was underestimated during the measurements. In addition to the underestimation of the dose rate, inaccurate measurement of the detector area can also lead to overestimation of sensitivity. The authors still have not explained exactly why this 1D perovskite can achieve such a surprisingly high sensitivity, which is the main reason that many reviewers have been skeptical about the sensitivity so far.

***Response 1:** We appreciate the reviewer's comments and his/her concern about the device sensitivity has already been replied to during the first round of revision. It should be noted that although the X-ray sensitivity of the 1D perovskite has reached the state-of-the-art level, higher sensitivity has already been reported (*Nat. Photonics* 2022, 16, 575). It is not surprising to achieve such a device sensitivity, since both the devices in the two works show similar $\mu\tau E$ product (or schubweg distance), which is proportional to the mean drift length of the charge carrier as the most important Figure-of-the-Merit to determine the charge collection efficiency/signal/sensitivity (Schlesinger, T. E. et al., *Materials Science and Engineering: R: Report* 2001, 32, 103-189; G.F. Knoll & D.S. McGregor, *MRS Online Proceedings Library*, 1993, 302, 3-17). Although the $\mu\tau$ product in the 1D $\text{TA}_4\text{Pb}_3\text{I}_{10}$ perovskite is relatively small, the $\mu\tau$ product only tells how fast the charge carriers can diffuse in the material. It is the $\mu\tau E$ product or schubweg distance that can determine the performance of a real device. This is why the amorphous-Se with a small $\mu\tau$ product can be commercialized.*

The high $\mu\tau E$ product of the 1D $\text{TA}_4\text{Pb}_3\text{I}_{10}$ perovskite is determined by the lattice structure, which can bear a large electric field and show good stability with a small dark current and current drift. Charges are also easily separated and transported in the crystals. As depicted in Figure 2a and Figure R1, the VBM of the 1D structure is contributed by the organic TA^+ ions, and the CBM is constructed by the inorganic octahedra part. The favorable energy level effectively suppresses the charge recombination, benefiting the charge carrier collection process. Besides, the strain-less F-1D detector exhibits extreme electric field stability, enabling stable high gain under high electric field intensity. It should be also noted that all the other reviewers are already satisfied with the response.

Figure R1: (a) is the VBM of the 1D TA₄Pb₃I₁₀. (b) is the CBM of the 1D TA₄Pb₃I₁₀.

Comment 2: The reversible transition between 1D and 2D perovskites is an interesting phenomenon, but the authors do not explain exactly why this transition can occur reversible many times. I agree with reviewer 3, during the transition, the chemical formulas changed a lot, which means a part of chemical did not exist in the final product. Although the author tries to explain this reason through thermal analysis, it is not convincing. Because part of the chemical component is lost after the sample is heated, the sample does not have the conditions for reversible phase transition.

Response 2: We agree with the reviewers' opinion on the chemical component change during the phase transition. As we claimed in the manuscript, the final F-1D tablet cannot be transitioned into 2D perovskite anymore after several times phase transitions.

The A-2D perovskite should go through the process of $3(TA_2PbI_4) = TA_4Pb_3I_{10} + 2(TAI)$. The excess TAI evaporates during thermal annealing or dissolves in the water, and the F-1D tablet finally stabilizes at the TA₄Pb₃I₁₀ structure. We follow the suggestion to add more discussion in the revised manuscript to make it clear.

Comment 3: In the revision, the authors measured the response time. However, limited information is given on the time response speed, as it is suggested that photoconductive gain and $u\tau E$ plays a significant role, it is necessary to comment its effect on the rise/fall time.

Response 3: Thanks for the reviewers' comments. The gain factor (G) can be described by:

$$G = \frac{\tau}{t} \quad (1)$$

where τ is the charge carrier lifetime, which determines the device response time. t is the charge carrier transit time, which can be calculated by:

$$t = \frac{d^2}{\mu \times V} = \frac{d}{\mu \times E} \quad (2)$$

Therefore, a large charge carrier mobility and electric field product can result in a small transit time and a large device gain, consistent with the 1D TA₄Pb₃I₁₀ perovskite performance. It should be noted that a large device gain doesn't require a very long charge carrier lifetime, since shallow traps can also lead to a large gain and fast response

speed (*Adv. Mater.* 2015, 27, 4975).

Comment 4: *In Figure S20, the authors obtained sensitivity by fitting the dose rate-dependent current-density of the detector. However, it seems that when the dose rate is 0, the detector still shows current density, which is very strange.*

Response 4: Thanks for the reviewers' comments. Firstly, we can clearly claim that when the dose rate is 0, the detector shows no signal current. Secondly, the linear variation exists within the linear dynamic range, we cannot assume that "when the dose rate is 0, the detector still shows current density" for the intercept of the fitting line is not zero. To further clarify the reviewers' confusion about the fitting line not passing the zero point, we also investigated some reported influential reference works where the intercepts of the fitting line are not zero as listed below:

- 1) Huang et al., *Nature Photonics* 2016, 10, 333-339
- 2) Yang et al., *Nature Photonics* 2019, 13, 602-608
- 3) Shen et al., *Nature Photonics* 2023, 16, 575-581
- 4) Chen et al., *Angew.* 2023, 62, e202302435
- 5) Huang et al., *Nature Communications* 2023, 12, 1686

Comment 5: *I agree with the reviewer 4, I would strongly recommend not reporting such spatial resolution from the single-pixel device, because it doesn't make sense in real X-ray imaging applications.*

Response 5: Thanks for the reviewers' comments. We follow the reviewer's suggestion to tune down our claim, but we believe it is still necessary to mention the spatial resolution. Since a high spatial resolution requires a very small pixel size and enough signal current. This further supports our claim on the high device sensitivity to meet the minimum signal value that the imaging system can stably capture and to distinguish the weak dose rate changes of the X-ray obscured by the image object. In our case, an X-ray image with high spatial resolution achieved by a small pixel can serve as good evidence. Although ignoring the current cross-talk between pixels, single-pixel scanning represents the comprehensive performance of a detector. It should be also noted that reviewer 4 is already satisfied with the response.

Reviewer 4

Comment: I appreciate the changes that the authors made to improve the manuscript. In my opinion, it might be published in the Nature Communication journal.

Response: We appreciate the reviewer's recognition of our work and the support for publication.

Reviewer 5

Comment 1: *The authors have addressed most of the comments from the previous review, however, I found some questions for ss-NMR data as below.*

In the ss-NMR data in Fig 3b, the author used the 2D $^{13}\text{C}\{^1\text{H}\}$ HETCOR to characterize the cross peaks and correlation between C and H to further confirm the structure, however, the author states that the solid-state ^1H spectra of both structures do not distinguish peaks of hydrogen atoms in different chemical environments. Thus, the HETCOR characterization for understanding which proton is attached to which C in a given molecule. However, it is not clear and cannot provide much useful information.

The author further used NOESY to understand the intermolecular interaction with water. It's interesting to see the chemical shift after titration with D_2O in ACN. However, why the authors change the D-solvent system from DMSO to ACN? From FigS12, the peaks in DMSO as solvent is much sharper than in ACN system. I suggest the authors also present the TAI- PbI_2 in DMOS with D_2O titration experiments to double-confirm the results. I think this part is much useful than 2D HETCOR data in Fig3b, I would suggest removing HETCOR data to SI and enlarging the NOESY data with details characterizations.

Response 1: Thanks for the reviewers' comments and suggestions.

1. As shown in Figure S12a, in a deuterated DMSO system, the chemical shift of the hydrogens on TAI- PbI_2 condition does not shift compared with pure TAI condition, representing the same monodisperse state of TAI molecules in both TAI- PbI_2 condition and TAI condition. However, in the deuterated ACN system, the chemical shift of the hydrogens on the TAI- PbI_2 condition shifts compared with the pure TAI condition, representing that intermolecular interactions occur between the TAI and PbI_2 . We think this is a transition state, and this state is much more suitable for D_2O titration experiments than the monodisperse state in DMSO.
2. We followed your suggestions to remove the HETCOR data to SI and enlarge the NOESY data. We also performed the TAI- PbI_2 in DMSO with D_2O titration experiments. However, as depicted in Figure R2, the peak intensity of hydrogen i and hydrogen d shows no obvious increase, meaning that the correlation between hydrogen i/d and hydrogen b shows no obvious change during the gradually adding D_2O into the DMSO system process. This is because of the excellent dispersity of TAI and PbI_2 in DMSO, it's too hard to drive TAI to interact with other TAI or PbI_2 in such a good solvent.

Figure R2: 1D selective NOESY spectra with titration of D₂O in deuterated DMSO.

Comment 2: In addition, the labeling in Figure 3b, 3c and figS12 are different but the discussion is on the TAI molecules, which is confusing when reading. I would suggest the author to keep them consistent in the discussion.

Response 2: Thanks a lot for the reviewer's comments and suggestions. We have followed your suggestions to unify the labels of hydrogens on TAI (Figure 3c and Figure S12). In 2D HETCOR data in Figure 3b, because the labels refer to carbons on TAI rather than hydrogens, we still labeled with Arabic numerals, but we have clearly claimed labels in the corresponding figure captions in *blue* in the revised manuscript.

REVIEWERS' COMMENTS

Reviewer #3 (Remarks to the Author):

The authors revised the manuscript and most of the issues were discussed. The paper is now good for publication.

Reviewer #5 (Remarks to the Author):

The authors answered and revised the MS to improve the quality of this work. Therefore, I would suggest publishing this work in Nature Communications.